# Subcellular spatial transcriptomics identifies three mechanistically different classes of localizing RNAs

Lucia Cassella [1,2] & Anne Ephrussi [1]

Intracellular RNA localization is a widespread and dynamic phenomenon that compartmentalizes gene expression and contributes to the functional polarization of cells. Thus far, mechanisms of RNA localization identified in *Drosophila* have been based on a few RNAs in different tissues, and a comprehensive mechanistic analysis of RNA localization in a single tissue is lacking. Here, by subcellular spatial transcriptomics we identify RNAs localized in the apical and basal domains of the columnar follicular epithelium (FE) and we analyze the mechanisms mediating their localization. Whereas the dynein/BicD/Egl machinery controls apical RNA localization, basally-targeted RNAs require kinesin-1 to overcome a default dynein-mediated transport. Moreover, a non-canonical, translation- and dynein-dependent mechanism mediates apical localization of a subgroup of dynein-activating adaptor-encoding RNAs (*BicD*, *Bsg25D*, *hook*). Altogether, our study identifies at least three mechanisms underlying RNA localization in the FE, and suggests a possible link between RNA localization and dynein/dynactin/adaptor complex formation in vivo.

RNA localization allows the precise compartmentalization of gene expression in space and time, and is a widespread phenomenon in many different cell types and organisms[1–5]. Three main mechanisms have been described to account for RNA localization: (1) active transport on cytoskeletal tracks, (2) localized protection from degradation, or (3) facilitated diffusion and entrapment[6]. Recently, several novel mechanisms have been reported to mediate RNA localization, such as hitch-hiking on other RNAs or organelles and co-translational RNA transport[7–15]. Active transport is the best-characterized mode of RNA localization and consists of the transport of ribonucleoprotein particles by motor proteins on cytoskeletal tracks. Localizing RNAs are typically transported in a translationally silent state and contain cis-acting localization elements (LEs) that are recognized and bound by trans-acting RNA-binding proteins (RBPs) mediating motor recruitment[16].

Kinesin motor proteins mostly mediate microtubule (MT) plus end-directed transport. Kinesin-1 (Khc) has been shown to mediate *oskar* (*osk*) RNA localization to the posterior pole of the *Drosophila* oocyte[17,18]. Whereas Tropomyosin-1 isoform I/C (atypical Tm1, *a*Tm1)

regulates *osk* posterior localization by directly stabilizing Khc interaction with the RNA[19–21], the Exon Junction Complex (EJC) deposited upon splicing is thought to activate kinesin-1 transport of the RNA[20]. Little is known about MT plus end-directed RNA transport in other tissues. Interestingly, *a*Tm1 is also important for *coracle* RNA localization at *Drosophila* neuromuscular junctions[22] and the EJC has been shown to mediate *NIN* RNA localization in human RPE1 cells[23].

Cytoplasmic dynein and its accessory complex dynactin direct trafficking of cargoes towards MT minus ends. In *Drosophila*, dynein-mediated RNA transport is accomplished by the dynein-activating adaptor Bicaudal-D (BicD) and the RNA binding protein Egalitarian (Egl)[24–26]. The dynein/BicD/Egl complex is thought to mediate nurse cell-to-oocyte transport of maternal RNAs, and was shown to direct apical RNA localization in the early embryo, neuroblasts, and polar cells[27–31]. The dynein/dynactin/BicD (DDB) motor complex is highly conserved and participates in the transport of different cargoes, with BicD (and its mammalian ortholog BICD2) linking the dynein motor to specific cargoes. While proteins binding to the BicD C-terminal domain (CTD), such as Egl or Rab6, impart cargo specificity[26,32–35], the BicD

[1]European Molecular Biology Laboratory, Meyerhofstrasse 1, Heidelberg 69117, Germany. [2]Collaboration for joint PhD degree between EMBL and Heidelberg University, Faculty of Biosciences, Heidelberg, Germany. ✉e-mail: anne.ephrussi@embl.org

N-terminal domain (corresponding to coiled-coil 1/2, CC1/2) binds to dynein/dynactin[33,36] and activates dynein processivity[26,37–39].

Although much of what is known about RNA localization comes from studies of maternally inherited RNAs in the *Drosophila* germline, several examples of localizing RNAs have been also reported in the follicular epithelium (FE) that envelops the germline cyst[4,40–44]. The FE is composed of highly polarized secretory follicle cells (FCs) belonging to the somatic lineage, with minus ends of non-centrosomal microtubules (ncMTs) anchored at the apical cell cortex facing the oocyte[45]. The FE is an easily manipulatable and powerful genetic system that, through the generation of mosaics, allows the dissection of the effect of mutations without disrupting developmental processes. Several lines of evidence indicate that the dynein/BicD/Egl RNA transport complex active in nurse cell-to-oocyte transport is also responsible for the apical localization of a handful of RNAs in the FE[31,40,42,46–48]. However, a comprehensive overview of RNA localization in the FE and its underlying mechanisms are lacking.

Here, we apply subcellular spatial transcriptomics to first identify the landscape of apically- and basally-localizing RNAs in the columnar FE. By screening a subset of apical and basal RNAs identified in this way, we find that the dynein/BicD/Egl machinery acts by default in directing apical RNA localization, and that an additional kinesin-1-dependent layer of regulation must be applied to direct basal RNA localization. Moreover, we identify a third, translation- and dynein-dependent mechanism that underlies the apical localization of transcripts encoding dynein-activating adaptors, providing a possible link between RNA localization and dynein/dynactin/adaptor complex formation in vivo.

## Results

### Identification of apical and basal RNAs in columnar follicle cells

To identify RNAs that localize apically or basally in *Drosophila* FE transcriptome-wide, we applied laser-capture microdissection (LCM) to isolate fragments of tissue that consisted of either the apical half ("apical domain") or basal half ("basal domain") of adjacent columnar follicle cells (Fig. 1a and Supplementary Movie 1). Differential gene expression analysis of apical vs. basal LCM-derived RNA-seq samples yielded 306 RNAs enriched in the apical samples and 249 RNAs enriched in the basal samples (false discovery rate [FDR] < 0.1) (Fig. 1b, Supplementary Data 1). Since LCM is highly susceptible to tissue contamination, we first aimed to identify those RNAs whose significant enrichment was a result of contamination by other cell types, such as the oocyte on the apical side or the circular muscles on the basal side (Supplementary Fig. 1a). To do so, we analyzed those RNAs characterized by high absolute log2-transformed fold change (|log2FC|) values of apical over basal abundance that might result from contamination of neighboring tissues expressing a different set of hallmark genes. By setting an arbitrary threshold of |log2FC| > 3 as indicative of contaminant RNA identity, we found 33 putative basal contaminants of muscle origin (log2FC < −3) and 2 putative apical contaminants of oocyte origin (log2FC > 3) (Fig. 1b and Supplementary Fig. 1b). 2/3 (*n* = 22) of basal genes with log2FC < −3 were annotated as being expressed or having a function in muscle tissues (FlyBase) and their mapped reads were often absent or in very low number in the apical fragments (Supplementary Fig. 1c, d). Moreover, we validated through single molecule Fluorescence In Situ Hybridization (smFISH) 3 putative basal contaminants (*Mhc*, *Act57B*, *wupA*) as being enriched in circular muscles with little or no expression in the FE (Supplementary Fig. 1e). This analysis resulted in 304 bona fide apical RNAs and 216 bona fide basal RNAs localizing in the columnar FE (Fig. 1b, Supplementary Data 1). Finally, 16 RNAs were randomly chosen from the computationally established list of significantly enriched bona fide apical or basal RNAs and were validated as true localizing RNAs through smFISH (Fig. 1c, d).

### Basal RNA localization depends on kinesin-1 and *a*Tm1

Basal RNA localization is a largely uncharacterized phenomenon. Previous reports have identified a limited number of basally-localizing RNAs in the FE[4,43,44], with little mechanistic insight. For this reason, we sought to elucidate the mechanisms behind basal RNA localization. Early reporter-based studies on the polarity of *Drosophila* tissues have shown that the basal domain of the FE is functionally similar to the posterior pole of the oocyte, as both compartments accumulate the MT plus end marker Kin:βgal[45]. Therefore, we hypothesized that the regulators of *oskar* posterior RNA transport might also be responsible for basal RNA localization in the FE. To test this hypothesis, we disrupted known components of the *osk* RNP transport machinery, such as kinesin-1 (Khc) and atypical Tropomyosin-1 (*a*Tm1) in the FE and analyzed the localization pattern of 4 validated basal RNAs (*Fkbp14*, *CG3308*, *Rtnl1*, *zip*) (Fig. 2). In all cells lacking either Khc (*Khc* RNAi cells) (Fig. 2a) or *a*Tm1 (*Tm1^NULL*,[21]) (Fig. 2b), basal RNA localization was severely disrupted, with all basal RNAs analyzed becoming apically localized. To have a quantitative overview of changes in RNA localization, we considered the ratio between the apical and the basal smFISH signal intensity in either wild-type (wt) or knock-down (KD) cells, and called this parameter Degree of Apicality (DoA), as values > 1 indicate an apical localization bias (Supplementary Fig. 2a). Then, we tested whether the DoA values of each RNA analyzed significantly differ in KD vs. wt cells by calculating the ratio between the DoA(KD) and the DoA(wt) for each RNA in each of the 3 conditions (see Materials and Methods and Fig. 2 for statistical testing). With this analysis, we confirmed that all basal RNAs analyzed were affected by lack of Khc or *a*Tm1 (Fig. 2d). To check whether the observed changes in RNA localization were specific to basal RNAs, we analyzed the localization pattern of four previously validated apical RNAs (*crb*, *msps*, *qtc*, *CG33129*) in the same mutant backgrounds. In contrast to basal RNAs, none of the apical RNAs analyzed was affected by disruption of kinesin-1-mediated RNA transport (Supplementary Fig. 3 and Fig. 2d), indicating that regulators of MT plus end-directed RNA transport specifically control basal RNA localization. These results show that kinesin-1 and *a*Tm1 are specifically responsible for basal RNA localization in the FE.

### Basal RNA localization depends on the EJC

In addition to Khc and *a*Tm1, nuclear events such as splicing and deposition of the Exon Junction Complex (EJC) play a role in the localization of *oskar* RNA at MT plus ends[49]. To test whether the EJC is involved in basal RNA localization, we disrupted the complex by overexpressing ΔC-Pym[50,51]. ΔC-Pym overexpression in FC clones impaired basal RNA localization (Fig. 2c–d), whereas apical RNA localization was unaffected (Supplementary Fig. 4a; Fig. 2d). This indicates that, in addition to Khc and *a*Tm1, the EJC is also involved in the regulation of basal RNA localization.

Surprisingly, basal RNAs are less represented than apical RNAs among RNAs previously shown to be preferentially bound by the EJC[52] (Supplementary Fig. 4b). Specifically, the basal RNAs we identified are on average EJC-depleted, while apical RNAs are on average EJC-enriched. This result is consistent with a slight apical enrichment of cytoplasmic GFP-Mago, as seen by GFP-Mago fluorescence quantification in the FE (Supplementary Fig. 4c–d). Since the EJC is specifically involved in basal RNA localization (Fig. 2c–d), these results were puzzling. However, considering that the EJC is displaced from RNAs upon translation[53], we further tested whether this discrepancy might be due to different translation rates of apical versus basal RNAs. To do so, we analyzed two different *Drosophila* ribo-seq datasets, from wild-type ovaries[54] and 0–2 h embryos[55] and extracted the values of translation efficiency of the apical and basal RNAs identified in our study. This analysis shows that basal RNAs are more translated than apical RNAs in both ovaries and 0–2 h embryos (Supplementary Fig. 4e), suggesting

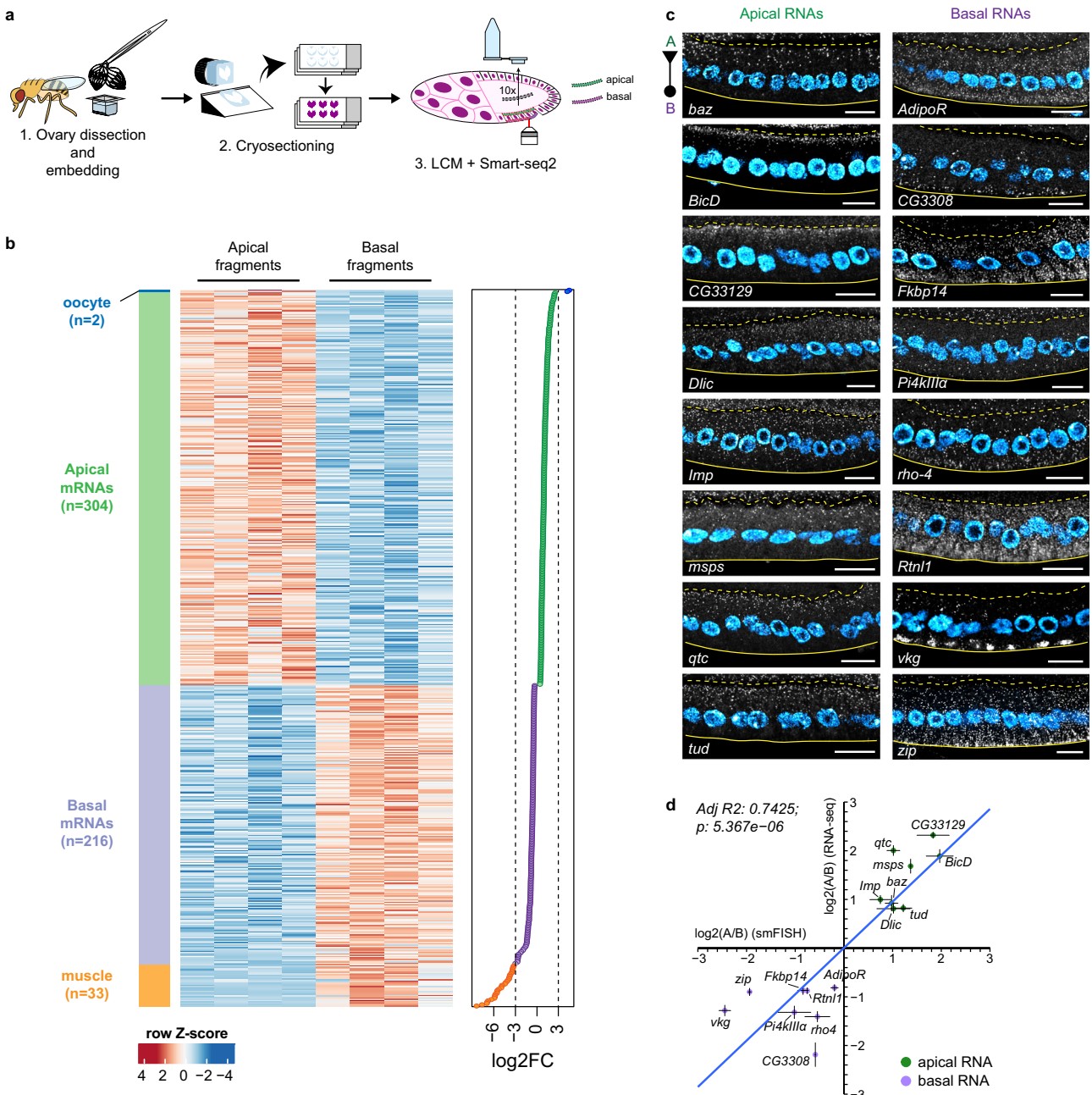

**Fig. 1 | Identification of apical and basal RNAs in *Drosophila* follicular epithelium by subcellular spatial transcriptomics. a** Schematic representation of the sample preparation procedure. **b** Heatmap representing RNA-seq signal (z-score of normalized read counts) for significantly enriched RNAs in microdissected apical and basal fragments (FDR < 0.1). Each row represents a significantly enriched RNA in either apical samples ($n = 4$) or basal samples ($n = 4$). The log2FC value of each RNA shown in the heatmap is indicated in the graph on the right. Dashed lines indicate threshold log2FC values (log2FC = −3 and log2FC = 3) arbitrarily set to identify oocyte contaminants (log2FC > 3, $n = 2$, blue), bona fide apical RNAs (0 < log2FC ≤ 3, $n = 304$, green), bona fide basal RNAs (−3 ≤ log2FC < 0, $n = 216$, purple), and muscle contaminants (log2FC < −3, $n = 33$, orange). **c** smFISH validation of 16 bona fide apical (left panels) and basal (right panels) RNAs. A dashed line and a continuous line in each panel delimit the FC-oocyte and FC-basal lamina borders respectively. A = apical domain (triangle); B = basal domain (circle). Nuclei (cyan) are stained with DAPI. Scale bars 10 μm. **d** Correlation of smFISH (x-axis) and RNA-seq (y-axis) apical vs. basal abundance (log2) for the 16 validated RNAs. The blue line corresponds to the fitted regression model ($n = 15$ degrees of freedom, coefficient: 0.95(0.14)). The adjusted R-squared and the p-value corresponding to the F-statistic are indicated in the graph. Each dot represents the average apical vs. basal fold change ± s.e.m. See also Supplementary Fig. 1, Supplementary Data 1 and Supplementary Movie 1. Source data are provided as a Source Data file.

that basal RNAs are on average EJC-depleted due to their higher translation rate.

Clonal analysis allows one to control for a change in the overall abundance of a given RNA in mutant cells by comparison of the total smFISH signal intensity with that in the adjacent wild-type cells. In both *Khc* RNAi and *ΔC-Pym* cells, the total smFISH signal intensity of basal RNAs was unchanged compared to neighboring wild-type cells,

showing that observed changes in RNA localization are not due to RNA degradation (Supplementary Fig. 2c).

## Mislocalization of the basal RNA *zip* in the absence of kinesin-1 depends on Egalitarian

Interestingly, disruption of MT plus end-directed RNA transport caused mislocalization of all analyzed basal RNAs to the apical domain.

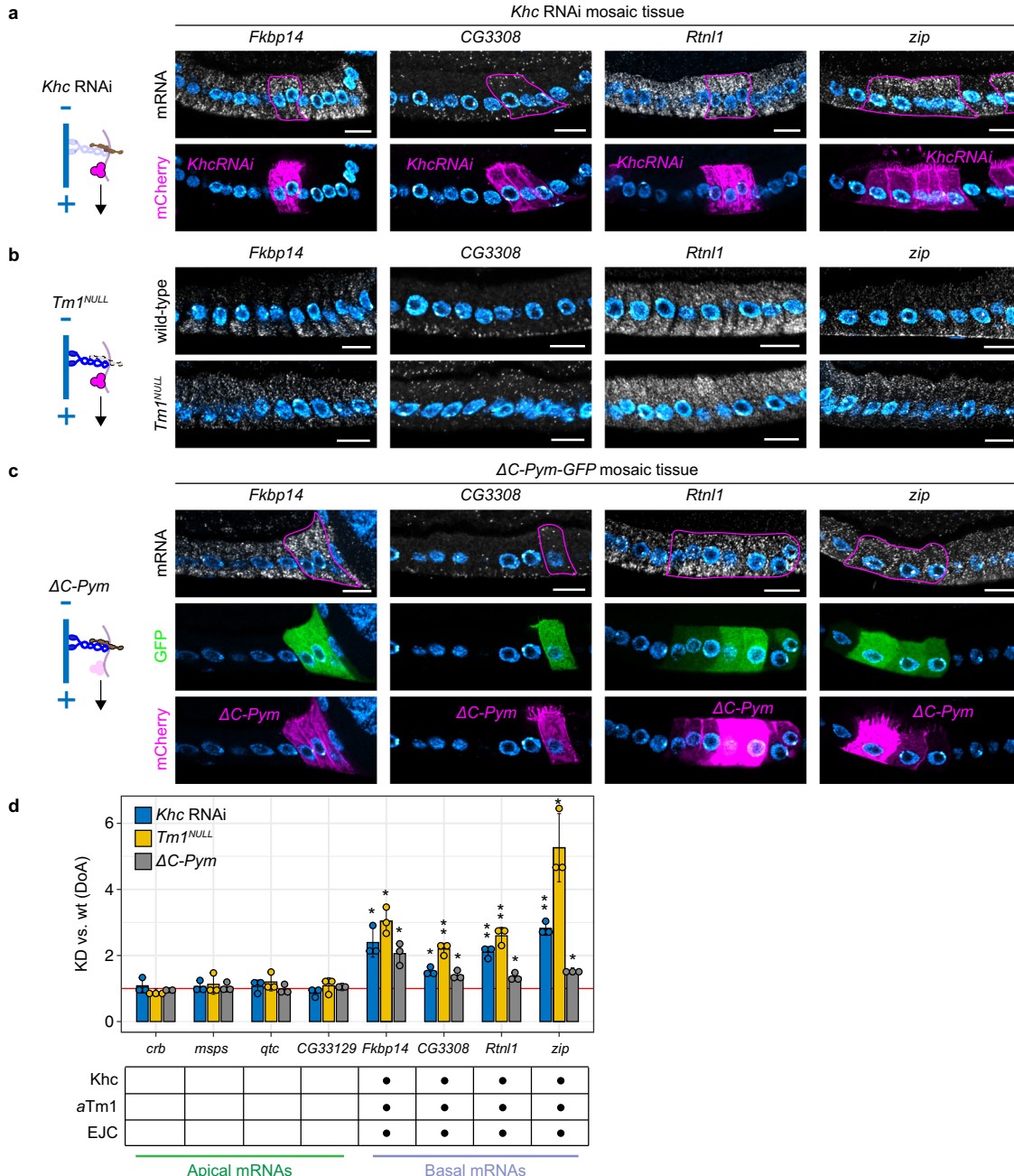

**Fig. 2 | Basal RNA localization depends on kinesin-1, *a*Tm1, and the EJC.** Mutant cells (marked with CD8-mCherry, lower panels) were generated by the UAS/Gal4 FLP-out system by inducing *Khc* RNAi (**a**) or by expressing the EJC-disrupting protein *ΔC-Pym* (**c**), to disrupt each component without significantly affecting tissue architecture. Neighboring wild-type cells are unmarked. A continuous line highlights mutant cells in smFISH images (upper panels). In **b** the expression of the *a*Tm1 isoform was specifically knocked down by generating *Tm1^{eg9}/Tm1^{egl}* (*Tm1^{NULL}*) egg chambers. **a** Localization of basal RNAs by smFISH in *Khc* RNAi mosaic tissue. **b** Localization of basal RNAs by smFISH in wild-type and *Tm1^{NULL}* egg chambers. **c** Localization of basal RNAs by smFISH in *ΔC-Pym-GFP* mosaic tissue. **d** Quantification of changes in the A-B distribution of apical and basal RNAs in conditions of downregulated kinesin-1 transport. Analyzed RNAs are indicated on the *x*-axis. The *y*-axis shows the average values (±2 SD) of the ratio between the Degree of Apicality (DoA) measured in knock-down (KD) cells and the DoA measured in wild-type (wt) cells for each RNA analyzed, in each of the three conditions. The mean KD/wt(DoA) value for each RNA in each condition was tested against a null hypothesis $H_0$ of KD/wt(DoA)=1 (red horizontal line), corresponding to no change between mutant and wild-type cells (one-sample two-sided *t*-test). Asterisks indicate mean values that significantly differ from the reference value of mu = 1 (*$p < 0.05$; **$p < 0.01$; ***$p < 0.001$; at least $n = 2$ biologically independent samples were analyzed for each gene in each of the three conditions). The table below the *x*-axis summarizes which RNAs were significantly affected by the lack of each regulator of kinesin-1-mediated transport. Nuclei (cyan) are stained with DAPI. Scale bars 10 μm. See also Supplementary Fig. 2, Supplementary Fig. 3, and Supplementary Fig. 4. Source data are provided as a Source Data file.

Several studies reported that apical RNA localization depends on the BicD/Egl machinery, a dynein-dependent complex that localizes RNAs apically in the blastoderm embryo and is thought to be responsible for nurse cell-to-oocyte transport of maternal RNAs. Therefore, the apical mislocalization of basal RNAs observed upon knock-down of kinesin-1 regulators might be due to apical RNA transport by the dynein/BicD/Egl machinery. To test this, we generated FC clones lacking either Egl (*egl* RNAi) or Khc (*Khc* RNAi), or both Egl and Khc [(*egl* + *Khc*) RNAi] and evaluated changes in the RNA localization of *zip*, one of the most striking examples of the apical mislocalization phenomenon (see

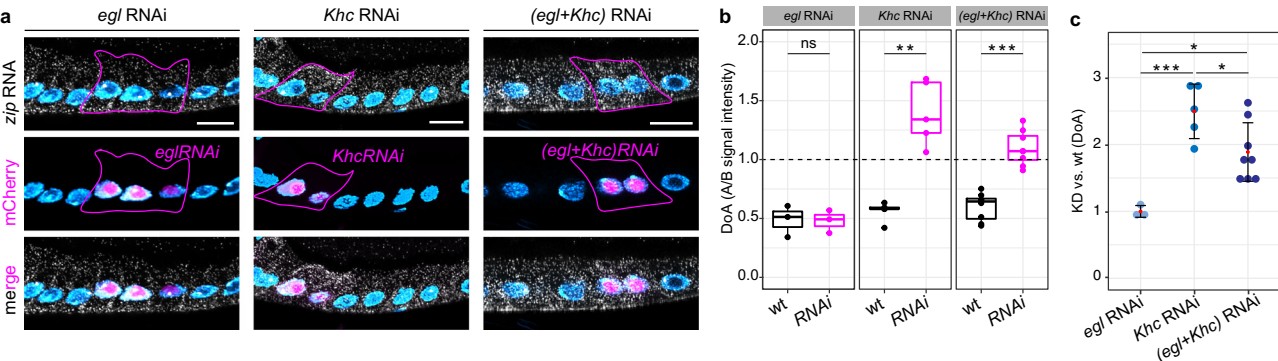

**Fig. 3 | Effects of double (*egl* + *Khc*) RNAi in *zip* RNA localization. a** *zip* RNA localization in *egl* RNAi, *Khc* RNAi and (*egl* + *Khc*) RNAi conditions visualized by smFISH. Mutant cells are marked by expression of mCherry in the nucleus (middle panels) and highlighted by a continuous line in smFISH images (upper panels). **b** Quantification of *zip* RNA signal (DoA) in wild-type (wt) and RNAi (*RNAi*) cells in *egl* RNAi, *Khc* RNAi and (*egl* + *Khc*) RNAi conditions. Values of DoA = 1 (dashed horizontal line) correspond to ubiquitous *zip* localization. Two-sided Student's *t* test was used to compare means. *egl* RNAi: $p = 0.94$ (ns); *Khc* RNAi: **$p = 0.00151$; (*egl* + *Khc*) RNAi: ***$p = 0.000004$. *egl* RNAi: $n = 3$; *Khc* RNAi: $n = 5$; (*egl* + *Khc*) RNAi:

$n = 8$. In boxplots, horizontal lines indicate the median, bounds of box define the first and third quartiles and the whiskers indicate ± 1.5 × interquartile range. **c** KD/wt change in *zip* DoA shows the variation of *zip* RNA localization in each experimental condition (mean ± 2 SD). Values close to 1 indicate no change in *zip* RNA localization upon RNAi. One-way ANOVA ($p = 0.00067$) followed by Tukey post-hoc tests were used to compare means. *egl* RNAi vs. *Khc* RNAi: ***$p = 0.0005$ ; *egl* RNAi vs. (*egl* + *Khc*) RNAi: *$p = 0.0141$ ; *Khc* RNAi vs. (*egl* + *Khc*) RNAi: *$p = 0.0444$ . *egl* RNAi: $n = 3$; *Khc* RNAi: $n = 5$; (*egl* + *Khc*) RNAi: $n = 8$. Nuclei (cyan) are stained with DAPI. Scale bars 10 μm. Source data are provided as a Source Data file.

Fig. 2a-c). Whereas *zip* RNA was unaffected upon *egl* RNAi and strongly apically mislocalized in *Khc* RNAi conditions as also highlighted by our previous experiments, (*egl* + *Khc*) RNAi caused *zip* to assume a ubiquitous localization that would be consistent with a failure of both kinesin-1-and dynein-mediated transport (Fig. 3a). *zip* DoA measurements in wt and *RNAi* cells in each of the three conditions provided a quantitative evaluation of the changes observed in smFISH experiments (Fig. 3b), with a significant decrease in KD/wt DoA in double (*egl* + *Khc*) RNAi cells (KD/wt DoA = 1.61) compared to *Khc* RNAi cells (KD/wt DoA = 2.49) (Fig. 3c). Therefore, despite being dispensable in basal RNA localization under normal conditions, the dynein/BicD/Egl complex is responsible for the apical mislocalization of a basal RNA (and possibly more) when kinesin-1 activity is lacking.

## Two different dynein-dependent mechanisms control apical RNA localization

As mentioned previously, several reports have identified the dynein/ BicD/Egl machinery as responsible for the apical localization of a subset of RNAs in the FE, such as *crumbs* (*crb*)[40,46]. To test in an unbiased way the degree of involvement of the dynein/BicD/Egl machinery in the localization of apical RNAs in the FE, we generated FC mutant clones in which either cytoplasmic dynein (*Dhc64C*, hereafter called *Dhc*) or Egalitarian (*egl*) were knocked-down by RNAi. We then analyzed the localization pattern of 5 validated apical RNAs (*crb, msps, qtc, CG33129, BicD*, with *crb* RNA as a positive control) by smFISH, along with the quantification of RNA localization by measuring the KD/wt DoA. The localization of all apical RNAs analyzed was completely abolished when *Dhc* was knocked down by RNAi, with the RNAs becoming ubiquitously distributed (Fig. 4a, c, d). *egl* RNAi caused all apical RNAs to lose their apical localization, with the surprising exception of *BicD* (Fig. 4b–d; see below). For the majority of apical RNAs analyzed, the total smFISH signal intensity was similar in mutant and wild-type cells (Supplementary Fig. 2d); only *CG33129* RNA displayed some degree of RNA degradation upon *Dhc* RNAi. In spite of this, *CG33129* signal was increased in the basal domain of RNAi cells with respect to wild-type cells, indicating that RNA degradation followed transcript mislocalization. Therefore, these results suggest that the changes observed in the localization of apical RNAs are not due to RNA degradation. In contrast to apical RNAs, basal RNAs largely maintained their basal localization pattern upon either *Dhc* RNAi or *egl* RNAi treatment (Supplementary Fig. 5a, b and Fig. 4d). Basal RNA

localization was only mildly affected in a subset of *Dhc* RNAi cells, likely as a consequence of the emergence of polarity defects in cells lacking Dhc[41,56](see Fig. 4a and Supplementary Fig. 5a).

The maintenance of *BicD* RNA localization in *egl* RNAi cells (Fig. 4c) was not due to a low efficiency of the RNAi, since both *egl* RNA and Egl protein were significantly reduced in *egl* KD cells (Supplementary Fig. 5c, d). Moreover, in egg chambers entirely lacking Egl throughout the FE (*egl*[NULL]FC, see Materials and Methods), *BicD* RNA was still apically localized, whereas localization of *CG33129* RNA, previously found to be Egl-dependent (see Fig. 4b), was disrupted (Supplementary Fig. 5e). Altogether, these results show that the dynein/ BicD/Egl complex is largely responsible for apical RNA localization, but a different dynein-dependent mechanism underlies the apical localization of *BicD* RNA. Considering that the Egl-independent targeting of *BicD* RNA represents a novel mechanism of apical RNA localization, we sought to gain more insight into the mechanisms regulating its RNA transport.

## *BicD* RNA localization requires an intact translation machinery

Localization of *BICD2/BicD* RNA at centrosomes in cultured cells is translation-dependent[57]. To test whether *BicD* RNA localization in the FE involves the same mechanism, we treated egg chambers ex vivo with the translation inhibitors puromycin (Puro) and cycloheximide (CHX) and analyzed the distribution of *BicD* RNA under these two conditions compared to control ovaries incubated in Schneider's medium only (Fig. 5a). To assess tissue integrity, in parallel we visualized *osk* RNA, whose localization during the middle stages of oogenesis should not be affected by translation inhibitors. Whereas the localization pattern of *BicD* RNA in CHX-treated egg chambers was similar to controls (Fig. 5b, d), Puro treatment clearly impaired *BicD* RNA localization in the FE (Fig. 5c). The distribution of *BicD* signal intensity along the A-B axis of mid-stage follicle cells shows that *BicD* enrichment at the apical cortex of the FE was severely reduced upon Puro treatment (Fig. 5e). As a proxy for the degree of signal mislocalization, we calculated the value corresponding to 50% of the cumulative area under the curve (a.u.c.) in Puro- or CHX-treated egg chambers and compared it with untreated controls. The results of this analysis show that the *BicD* RNA signal shifted significantly towards the basal domain in Puro-treated egg chambers, whereas CHX had no effect on *BicD* RNA localization (Fig. 5f). The fact that freezing elongating ribosomes (CHX condition) does not affect *BicD* RNA

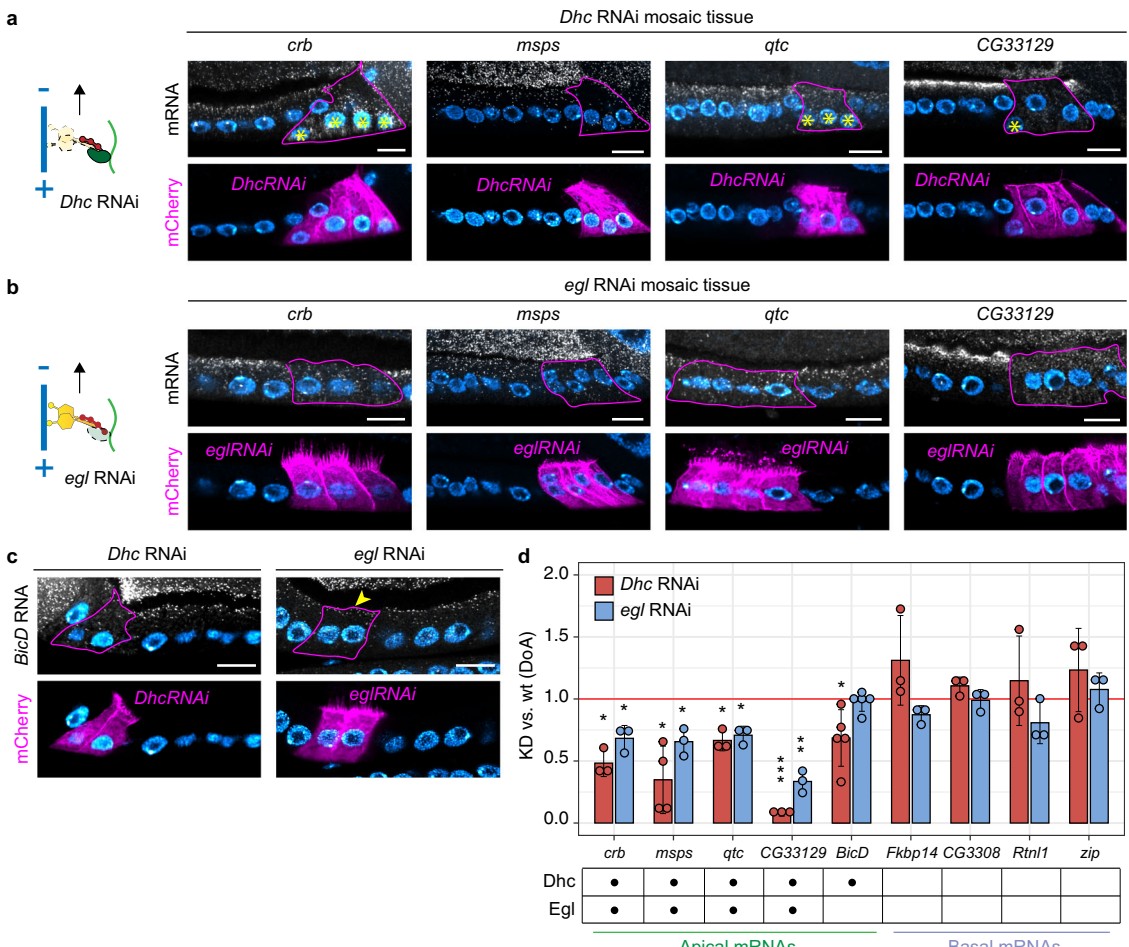

**Fig. 4 | Two different dynein-dependent mechanisms control apical RNA localization.** Localization of apical RNAs by smFISH in *Dhc* RNAi (**a**) and *egl* RNAi mosaic tissue (**b**). Mutant cells are marked by the expression of CD8-mCherry (lower panels) and highlighted with a continuous line in smFISH images (upper panels). Neighboring wild-type cells are unmarked. Asterisks (*) indicate basal mispositioning of nuclei due to *Dhc* RNAi, an indication of mild cell polarity defects. **c** *BicD* RNA localization in *Dhc* RNAi (left) and *egl* RNAi (right) mosaic tissues. The arrowhead indicates the apical persistence of *BicD* RNA in *egl* RNAi cells. **d** Quantification of changes in the A-B distribution of apical and basal RNAs in conditions of downregulated dynein/BicD/Egl transport (*Dhc* RNAi or *egl* RNAi). Analyzed RNAs are indicated on the *x*-axis. The *y*-axis shows the average values

($\pm 2$ SD) of the KD/wt ratio (DoA) for each RNA analyzed, in each of the two conditions. The mean KD/wt(DoA) value for each RNA in each condition was tested against a value of KD/wt(DoA)=1 (red horizontal line), corresponding to no change between mutant and wild-type cells (one-sample two-sided t-test). Asterisks indicate mean values that significantly differ from the reference value of mu=1 (*$p < 0.05$; **$p < 0.01$; ***$p < 0.001$; at least $n = 3$ biologically independent samples were analyzed for each gene in each of the three conditions). The table below the x-axis summarizes which RNAs were significantly affected by the lack of each regulator of dynein-mediated transport. Nuclei (cyan) are stained with DAPI. Scale bars 10 μm. See also Supplementary Fig. 2 and Supplementary Fig. 5. Source data are provided as a Source Data file.

localization, whereas blocking translation by releasing the nascent peptide (Puro condition) does, indicates that an intact translation machinery and the presence of a nascent peptide may be required for *BicD* RNA localization in FCs.

## *BicD* RNA is co-translationally localized

To understand whether the localization of *BicD* depends on translation of its own RNA (in *cis*) or of other factors (in *trans*), we designed a series of transgenic constructs consisting of a BicD-GFP cassette inserted downstream of an 18-bp linker in which we could introduce the desired frameshift mutations without disrupting any unknown RNA localization element in the BicD CDS (Supplementary Fig. 6a). Each of these transgenes was expressed in FC clones in a *BicD* wild-type background and the transgenic *BicD-GFP* RNA was specifically detected by smFISH using antisense GFP probes. *GFP* RNA carrying the same 3′ untranslated region (UTR) as BicD-GFP constructs failed to localize when expressed in the germline or in the FE (Supplementary Fig. 6b), showing that this sequence alone is not sufficient to drive RNA localization. In contrast, $^{o}BicD$-GFP RNA

("*In-frame*") showed a strong apical localization in FCs (Fig. 5g–i), similarly to the endogenous *BicD* RNA (see Supplementary Fig. 7a). Moreover, the expression of full-length BicD-GFP was validated by the presence of GFP fluorescence in CD8-mCherry[+] cells expressing the transgene (Fig. 5g). Disruption of the BicD-GFP reading frame by either +1 or −1 frameshift ("*Frameshift*"), verified by the absence of GFP signal in CD8-mCherry[+] cells, was sufficient to impair apical RNA localization (Fig. 5g–i). Importantly, the change in localization observed for *Frameshift* BicD-GFP RNAs was not due to RNA degradation, as the smFISH signal intensity of all RNA constructs was comparable (Fig. 5j). Consistent with the puromycin-induced impairment of RNA localization in the FE, these results show that *BicD* RNA is co-translationally localized at the apical cortex.

## *BicD* and *Dhc* RNAs decorate dynein particles at the apical cortex

As in BicD the first peptide emerging from the ribosome is the dynein-binding domain[33], the co-translational localization of *BicD* RNA might

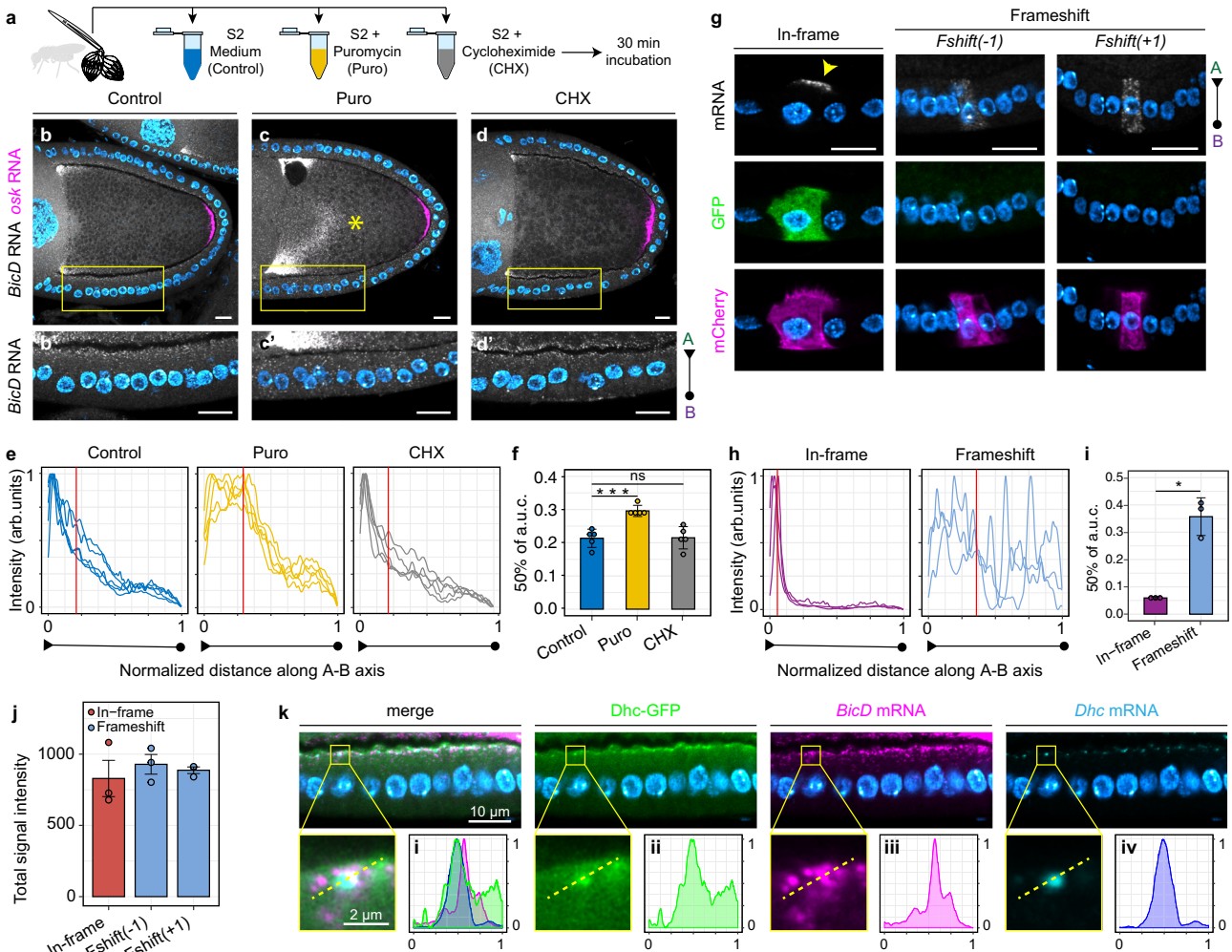

**Fig. 5 | *BicD* RNA is co-translationally localized at cortical *Dhc* foci. a** Schematic of the ex vivo translation inhibitors treatment. **b−d** Dual-color *BicD* (grayscale) and *osk* (magenta) smFISH experiments on Control (b-b'), Puro- (c-c') or CHX-treated (d-d') ovaries, with a magnification of the follicular epithelium (bottom panels). Note the mislocalization of *BicD* RNA towards the center of the oocyte (*) in **c. e−f** *BicD* RNA distribution along a linear ROI spanning the apical-basal axis of follicle cells measured as smFISH fluorescence intensity in the three conditions. A red vertical line represents the mean x value corresponding to the 50% of the cumulative area under the curve (a.u.c.), a proxy for *BicD* localization. Apical (triangle) to basal (circle) directionality of signal measurement is indicated below each graph. The bar-plot in **f** shows mean a.u.c. (±2 SD, *n* = 5). Control-Puro: ***$p$ = 0.000879 ; Control-CHX: $p$ = 0.9374 (ns). **g** Expression of different BicD-GFP constructs (In-frame: $^{0}$BicD-GFP; Frameshift: $^{(-1)}$BicD-GFP and $^{(+1)}$BicD-GFP) in FC clones and analysis of transgenic RNA distribution by smFISH with antisense GFP probes. CD8-mCherry (magenta) marks clones expressing each BicD-GFP construct. **h, i** *In-frame* (purple)

and *Frameshift* (light blue) *BicD-GFP* RNA distribution (**h**) and mean a.u.c.(±2 SD, *n* = 3) (**i**). *$p$ = 0.01719 . **j** Total smFISH signal (apical + basal, arb. units) of the three *BicD-GFP* RNA constructs analyzed (± s.e.m) (*n* = 3). In-frame vs. $^{(-1)}$BicD-GFP: $p$ = 0.5406 (n.s); In-frame vs. $^{(+1)}$BicD-GFP: $p$=0.6966 (n.s). **k** Localization of *BicD* RNA (magenta), *Dhc* RNA (cyan), and Dhc-GFP (green) in stage 10 FE. Insets show a magnification of a single Dhc-GFP/*Dhc* RNA focus. A dashed line indicates the cross-section along which each signal was measured (panels i−iv). Signal intensities (y-axis) and line length (x-axis) were normalized in the 0−1 range. Notably, *Dhc* RNA was not present in our RNA-seq list of apically-enriched RNAs, likely due to the fact that, to minimize contamination from the adjacent tissues, we performed laser cutting excluding cortical regions of the FE where *Dhc* RNA localizes. *P*-values were estimated by Welch two-sample two-sided *t*-test. Nuclei (cyan) are stained with DAPI. Scale bars 10 µm unless otherwise specified. See also Supplementary Fig. 6 and Supplementary Fig. 7. Source data are provided as a Source Data file.

depend on association of nascent BicD protein with dynein. To have an indication whether this might be the case, we imaged *BicD* RNA by smFISH in ovaries expressing endogenously tagged Dhc-GFP[58]. Furthermore, hypothesizing that nascent BicD might assemble co-translationally with nascent dynein, we included smFISH probes to also detect *Dhc* RNA. Although the Dhc-GFP signal was diffuse in the ovary, distinct Dhc-GFP foci were detected at the apical cortex of columnar FCs (Fig. 5k) and elsewhere in the germline (see below). These foci also contain *Dhc* RNA, indicating that these might be sites of *Dhc* RNA translation. *BicD* RNA showed a partial co-localization with Dhc-GFP/*Dhc* RNA foci, consistent with the hypothesis of its co-translational association with newly synthesized Dhc protein at the apical cortex.

## The first step of *BicD* RNA localization in the early cyst is translation-independent

In the germline, BicD has an instructive role in oocyte specification[24,59,60]. Importantly, *BicD* RNA localization reflects MT minus end enrichment[45,61] in both the germline and FE (Supplementary Fig. 7a). We noticed that, as in the FE, *BicD* RNA localization to the anterior of the oocyte (stages 9–10) was impaired in Puro-treated ovaries, whereas CHX had no effect (Fig. 5c, d). Interestingly, Puro treatment impaired the dynein-dependent posterior localization of *BicD* RNA in early stage oocytes (stages 4-5), without significantly affecting nurse cell-to-oocyte transport, a process that is also dynein-dependent (Supplementary Fig. 7b). Similarly, germline-expressed *Frameshift* BicD-GFP RNA enriched in the oocyte, but lost the posterior

localization displayed by both endogenous *BicD* RNA and *In-frame* BicD-GFP RNA (Supplementary Fig. 7c). In addition, we found that *BicD* RNA decorates Dhc/*Dhc* RNA foci in both the FE and the oocyte, but not in the nurse cells (Supplementary Fig. 7d). Taken together, these results indicate that the mid-oogenesis oocyte and the columnar FCs share a similar co-translational mechanism for *BicD* RNA localization. In contrast, *BicD* RNA nurse cell-to-oocyte localization appears to be mediated by a translation-independent mechanism that does not involve the association with Dhc/*Dhc* RNA particles.

### A subset of dynein adaptor-encoding RNAs are also co-translationally localized in the FE

BicD belongs to the class of dynein-activating adaptors, linking cargoes to the dynein motor complex[62]. We found that the RNA encoding all *Drosophila* orthologs of the currently known or putative dynein activating adaptors (hereafter collectively called "adaptor-encoding RNAs"), namely *hook* (HOOK1-3), *Bsg25D* (NIN/NINL), *Nuf* (RAB11FIP3), and *Milton* (TRAK1-2), were significantly enriched apically in our list of localizing transcripts (Supplementary Data 1), with the exception of *Spindly* (SPDL1) which was below the detection threshold. By hypothesizing that the same dynein-dependent co-translational process that drives *BicD* RNA localization would also be responsible for the apical localization of adaptor-encoding RNAs, we tested whether the localization of adaptor-encoding RNAs was affected by either *Dhc* or *egl* RNAi. Interestingly, the apical localization of *Bsg25D* and *hook* was significantly disrupted in *Dhc* RNAi cells (Fig. 6a, c), but not in *egl* RNAi cells (Fig. 6b, c). Moreover, the apical localization of both adaptor-encoding RNAs showed sensitivity to Puro but not CHX, as seen by smFISH (Fig. 6d). In particular, apical-cortical *Bsg25D* signal was lost upon Puro treatment, with the signal re-distributing more basally. *hook*, which appeared more abundant than *Bsg25D*, became unlocalized upon Puro treatment. As *Bsg25D* smFISH signal was characterized by a low signal-to-noise ratio, we analyzed the effect of translational inhibitors on the localization of the adaptor-encoding RNAs by measuring the integrated density after signal thresholding in three subdomains: apical-cortical (apical-most 1/3 of the apical area of FCs), subapical (the remaining 2/3), and basal (Supplementary Fig. 2b). Then, we calculated the percentage of signal in each subdomain, for each RNA (including *BicD*) in Control, Puro, and CHX conditions. This analysis revealed that, similar to *BicD* RNA, the localization of *hook* and *Bsg25D* RNAs was significantly affected by Puro treatment, whereas CHX had no effect (Fig. 6e).

To further investigate whether *hook* and *Bsg25D* use the same localization mechanism as *BicD*, we analyzed their spatial relationship with Dhc-GFP/*Dhc* RNA particles. As for *BicD*, both *Bsg25D* (Fig. 6f) and *hook* (Fig. 6g) were shown to partially co-localize with, thus decorate, Dhc-GFP foci containing *Dhc* RNA. Overall, these results suggest that the RNAs encoding the dynein activating adaptors *BicD*, *hook*, and *Bsg25D* represent a subgroup of apical RNAs that share the same co-translational, dynein-dependent mechanism that ensures their localization at cortical dynein foci also containing *Dhc* RNA.

## Discussion

Only few examples of localizing RNAs in the FE have been described to date, with little mechanistic insight[4,40–44]. To explore the extent of RNA localization in a somatic tissue in vivo and gain insight into the mechanisms underlying the phenomenon, we have used laser-capture microdissection of apical and basal subcellular fragments of columnar follicle cells coupled with RNA-seq to identify localizing RNAs in this tissue. This allowed us to investigate in detail the landscape of mechanisms that mediate both apical and basal RNA localization in the FE (Fig. 7a). In our study, we found that basal RNA localization is mechanistically analogous to posterior RNA localization in the oocyte (represented by *osk*), reflecting MT plus end enrichment[45]. Khc, *a*Tm1, and the EJC appear to be core components of a general basal RNA

localization machinery. These results are in line with previous findings on *osk* RNA indicating that Khc/*a*Tm1 bind to the 3'UTR[20] and the EJC activates kinesin-1 transport through association with the coding sequence[63].

According to our analysis, deposition of the EJC is necessary but not sufficient to determine RNA localization, as we found that the EJC is deposited on both apically- and basally-directed RNAs. Interestingly, Kwon et al.[23] found that the EJC specifically localizes to the basal body of the primary cilium in mono-ciliated cells, where it controls the centrosomal localization of *NIN* RNA towards MT minus ends. In contrast, our data suggest that in columnar follicle cells, which are characterized by non-centrosomal MTOCs[64–66], the EJC may play a role in MT plus end-directed (basal) RNA transport by acting synergistically with kinesin-1 and *a*Tm1. Strikingly, the localization of *osk* RNA to the posterior pole of the oocyte also relies on the presence of MTs generated from non-centrosomal MTOCs[65]. Therefore, the mammalian EJC might have acquired a specific role in the localization of *NIN* RNA at basal bodies of mono-ciliated cells, while the *Drosophila* EJC appears to contribute to the MT plus end-directed localization of several RNAs through a centrosome-independent mechanism both in the somatic follicular epithelium (basal RNAs) and in the germline (*osk* RNA).

Interestingly, when either component of the kinesin-1 transport complex was lacking, basal RNAs were mislocalized to the apical domain in a dynein-dependent process. Therefore, dynein-mediated apical localization represents a default mechanism that must be overcome by kinesin-1 to drive basal RNA localization. Two possible scenarios could explain dynein-mediated apical mislocalization upon kinesin inhibition. Dynein and kinesin-1 could be engaged in a tug-of-war, pulling the RNAs in opposing directions, a phenomenon observed in the transport of vesicles and lipid droplets[67]. Alternatively, the dynein complex could be kept in an inhibited state and activated upon disruption of kinesin-1 and its regulators. If the tug-of-war scenario were correct, we would have expected a change in *zip* RNA localization in all RNAi conditions including *egl* RNAi alone, namely a shift to a more basal localization due to the enhanced Khc-dependent motility. However, since we did not see a significant change in *zip* localization when only Egl was knocked down, the tug-of-war hypothesis appears to be less likely than the inhibition hypothesis. In addition, this phenomenon recalls *osk* RNA mislocalization to the oocyte anterior upon disruption of kinesin-1, *a*Tm1 or EJC components[17,18,21,68–72] which was hypothesized to occur due to a failure to inactivate dynein-mediated RNA transport[18].

Apical RNA localization, on the other hand, can be divided into two mechanistically distinct categories, both based on dynein-mediated transport. The first category includes those RNAs that are transported apically by the dynein/BicD/Egl machinery, a well characterized RNA transport complex that directs RNAs towards MT minus ends in a variety of tissues[29]. Our data suggest that the majority of apically localizing RNAs may belong to this class, as the localization of most of our randomly chosen apical RNAs was affected in both *Dhc* RNAi and *egl* RNAi conditions. This hypothesis is consistent with previous studies that identified several apical RNAs as BicD/Egl cargoes, in a variety of *Drosophila* tissues[31,40,42,46–48]. The BicD/Egl machinery has been hypothesized to be part of a larger RNP complex that ensures a tight translational control of the transported RNA[73]. Our finding that basal RNAs are on average more translated than apical RNAs suggests that RNAs transported apically in the FE by the dynein/BicD/Egl transport complex might indeed be kept in a translationally silent state until they have reached their final destination.

The second category of dynein-dependent apical RNAs does not involve Egalitarian activity for their localization. This includes a subgroup of dynein-activating adaptors, namely *BicD, hook*, and *Bsg25D* (BICD2, HOOK1-3, and NIN/NINL in mammals). Common features of their apical RNA localization include sensitivity to puromycin and partial co-localization with cortical dynein foci containing also *Dhc*

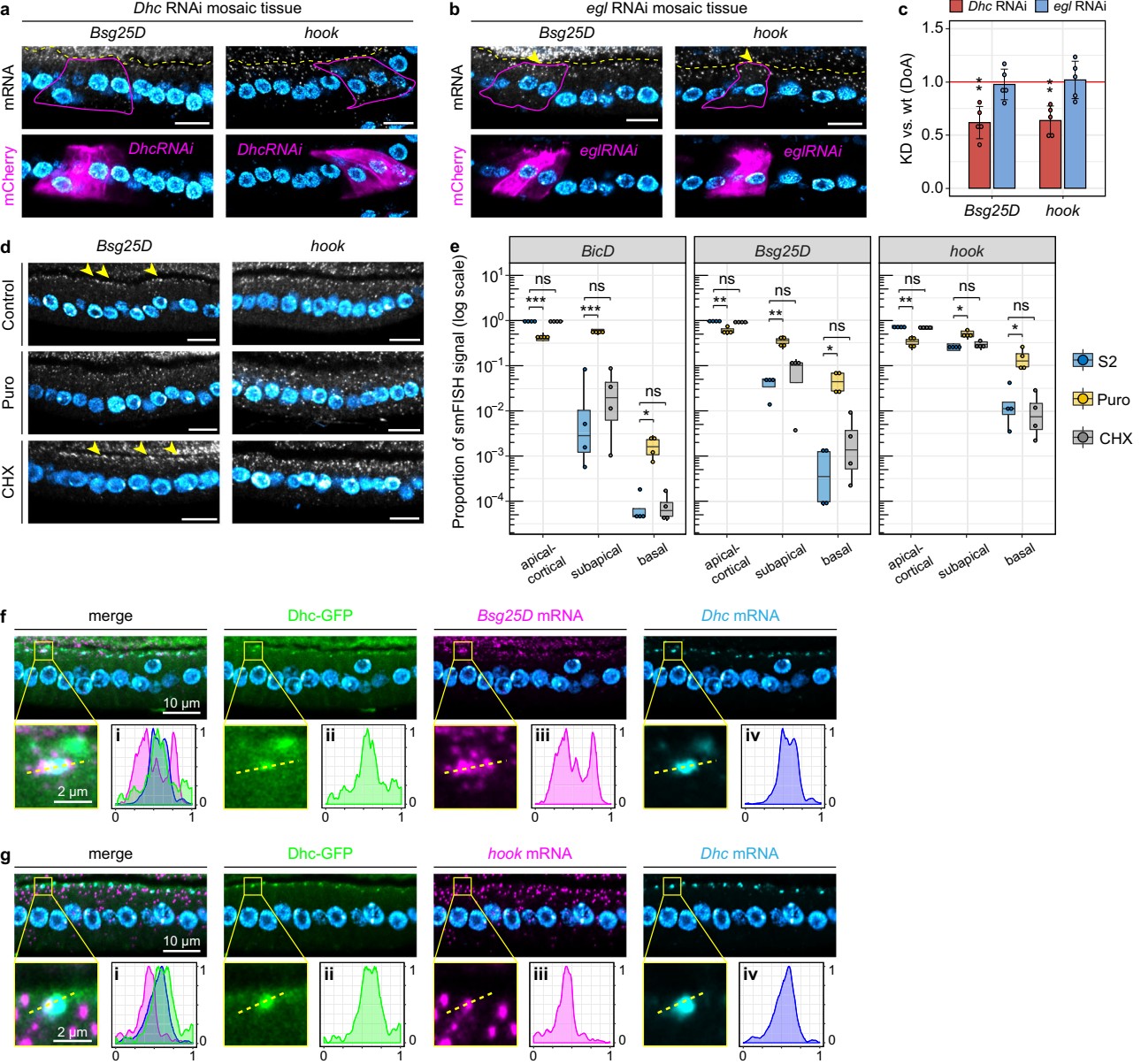

**Fig. 6 | The apical localization of *hook* and *Bsg25D* RNAs is mechanistically similar to *BicD*.** Localization of *Bsg25D* and *hook* RNAs by smFISH in *Dhc* RNAi (**a**) and *egl* RNAi mosaic tissue (**b**). Mutant cells are marked by CD8-mCherry (lower panels) and highlighted with a continuous line in smFISH images (upper panels). Neighboring wild-type cells are unmarked. Arrowheads in **b** indicate the persistence of *Bsg25D* and *hook* RNAs apically in *egl* RNAi cells. A dashed yellow line demarcates the oocyte-FE border. **c** A-B distribution of *Bsg25D* and *hook* RNAs in conditions of downregulated dynein/BicD/Egl transport. The *y*-axis shows the average values (± 2 SD) of the KD/wt DoA for each RNA analyzed, in each condition. The mean KD/wt DoA value for each RNA in each condition was tested against a value of KD/wt(DoA)=1 (red horizontal line, one-sample two-sided *t*-test). Asterisks indicate mean values that significantly differ from the reference of mu=1 (*$p < 0.05$; **$p < 0.01$; ***$p < 0.001$; $n = 5$). **d** *Bsg25D* (left panels) and *hook* (right panels) RNA localization in the FE visualized by smFISH in Control, Puro- or CHX-treated ovaries. Arrowheads indicate prominent *Bsg25D* apical-cortical foci in

Control and CHX conditions that disappear upon Puro treatment. **e** *BicD* (left), *Bsg25D* (middle), and *hook* (right) RNA distribution along the A-B axis of FCs measured as smFISH fluorescence intensity in Control (blue), Puro (yellow) and CHX (grey) conditions. The *y*-axis shows the percentage of smFISH signal (measured as integrated signal density) in each sub-domain. Differences in RNA localization were estimated by two-sided *t*-test (ns = not sig; *$p < 0.05$; **$p < 0.01$; ***$p < 0.001$; $n = 4$). Boxplots show the median (horizontal line), the first and third quartiles (bounds of box), ±1.5 × interquartile range (whiskers). Localization of *Bsg25D* RNA (**f**) or *hook* RNA (**g**) (magenta), *Dhc* RNA (cyan), and Dhc-GFP (green) in stage 10 FE. Insets show a magnification of a single Dhc-GFP/*Dhc* RNA focus. A dashed line indicates the cross-section along which each signal has been measured (panels i–iv). Signal intensities (*y*-axis) and line length (*x*-axis) were normalized in the 0–1 range. Nuclei (cyan) are stained with DAPI. Scale bars 10 μm unless otherwise specified. See also Supplementary Fig. 2. Source data are provided as a Source Data file.

RNA. Moreover, both *Bsg25D*[74] and *BicD* (this study) RNA constructs containing the CDS alone are sufficient for the accumulation of their encoded protein at MT minus ends. Puromycin causes the disassembly of the translational machinery and the release of the N-terminal peptides emerging from ribosomes. As the N-terminal portion of these adaptors binds dynein or dynactin subunits[33,75–78], we propose that the

apical localization of *BicD, hook*, and *Bsg25D* depends on the co-translational association between dynein components and nascent adaptors at cortical dynein foci (Fig. 7b). This process might also be conserved in mammals, since the localization of both *BICD2* and *NIN* RNA was shown to be puromycin-sensitive[57]. Previous studies have shown that the presence of either BICD2, HOOK3 or NIN/NINL

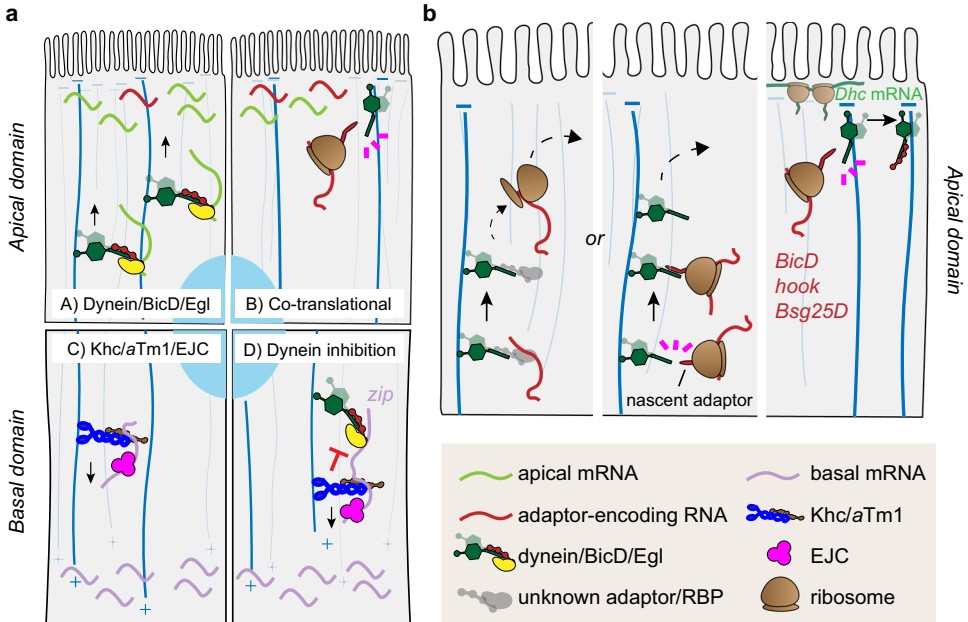

**Fig. 7 | Models of RNA localization mechanisms in follicle cells. a** Model of the mechanism underlying apical (A-B) and basal (C-D) RNA localization. Apical RNAs are localized at MT minus ends by two dynein-dependent mechanisms: (A) the dynein/BicD/Egl RNA transport machinery localizes most of apical RNAs in the FE; (B) a subset of dynein adaptor-encoding RNAs (*BicD*, *Bsg25D*, and *hook*) localize co-translationally at cortical dynein foci. (C) Basal RNAs are localized by Khc/*a*Tm1/EJC moving towards MT plus ends enriched basally. (D) In the transport of basal RNAs, the dynein complex is kept in an inhibited state by kinesin-1 and its regulators.

**b** Model for the apical localization of adaptor-encoding RNAs. RNAs can reach the apical domain by either canonical RNA transport by an unknown RBP complex or by interaction of the nascent adaptor protein with dynein/dynactin transporting the translationally engaged RNA to the apical domain. Once at the apical cortex, the nascent adaptor associates through its N-terminal domain with newly translated cortically-anchored dynein, presumably allowing the relief of both proteins' autoinhibition.

promotes the formation of highly processive dynein/dynactin complexes[37,38,79]. Therefore, it is possible that co-translational assembly of components of the dynein-adaptor complexes is necessary to overcome dynein auto-inhibition[80,81]. *BicD*, *hook*, and *Bsg25D* may co-translationally associate with dynein soon after nuclear export of the RNA, promoting its apical transport in a manner similar to what has been proposed for *PCNT* RNA targeting at centrosomes[14]. Alternatively, since dynein can also function as a MT-tethered static anchor in mid-oogenesis oocytes and follicle cells[82,83], the interaction between dynein and nascent adaptor proteins could occur after the RNA has reached the cell cortex by dynein-mediated transport. Indeed, puromycin treatment did not completely abolish the apical enrichment of adaptor-encoding RNAs, despite causing a marked decrease in their signal close to the apical cortex, where they decorate dynein cortical foci.

In vitro studies have shown that full-length BicD/BICD2 adopts an autoinhibitory conformation resulting from CC1/2 folding onto the CTD-containing CC3[26,36,84]. Although the leading hypothesis in the field is that cargo binding to the CTD is responsible for the alleviation of auto-inhibition by freeing up the N-terminal dynein-binding domain[26,32,33,36], it is possible that in vivo both nascent BicD interaction with dynein and cargo binding to the CTD might cooperate in preventing BicD intramolecular inhibition in the cellular environment. Strikingly, whereas the mechanism underlying oocyte localization of *BicD* RNA during mid-oogenesis resembles that observed in follicle cells, the nurse cell-to-oocyte transport of *BicD* RNA appears to be governed by a different, translation-independent mechanism that may not involve interaction with Dhc/*Dhc* RNA particles, consistent with a previous study indicating that *BicD* RNA is translationally inhibited by Me31B in the nurse cells[85]. In contrast to early egg chambers in which the MT network emanates from a posteriorly-positioned microtubule organizing center in the oocyte, mid-stage oocytes and columnar follicle cells are both characterized by non-centrosomal MTs (ncMTs)

tethered to the cell cortex[86]. Therefore, the establishment of ncMTs could be at the basis of the mechanistic switch from translation-independent to co-translational *BicD* RNA localization in these compartments. A recent report has shown that *NIN* RNA (the mammalian ortholog of *Bsg25D*) localizes at ncMTs and its expression is essential for apico-basal MT formation and columnar epithelial shape[87]. Therefore, it is possible that the co-translational transport of adaptor-encoding RNAs may be important for correct ncMT nucleation at the apical cortex of the follicular epithelium.

## Methods

### LCM sample preparation
*w1118* virgin females were kept with males for 24 h at 25 °C on yeast-supplemented cornmeal food.

Ovaries were dissected in PBS, transferred to a cryomold and snap-frozen in cold 2-Methylbutane after removal of excess PBS. Frozen ovaries were immediately covered with OCT cryoembedding compound (Sakura) and snap-frozen again. Before cryostat sectioning, each block was equilibrated at −20 °C for 1 h. 10 μm cryosections of OCT-embedded ovaries were carefully placed on a MembraneSlide NF 1.0 PEN (Zeiss), briefly thawed at RT and immediately fixed in 75% RNase-free (RF) ethanol for 30 s. Excess OCT was removed with ddH2O RF, and slides were stained in 100 μl Histogene staining solution (Arcturus) according to the manufacturer's instructions. Finally, sections were dehydrated in increasing ethanol concentrations (75%, 95%, 100%), and briefly air-dried before LCM.

### LCM and RNA-seq
LCM was performed with a Zeiss PALM MicroBeam and visualized under a 63X objective. Sectioned mid-oogenesis egg chambers were staged according to morphological criteria. Once stage 9–10 egg chambers had been identified, either the apical half ("apical fragment") or the basal half ("basal fragment") of 5-10 contiguous columnar follicle

cells was microdissected and collected into the cap of an AdhesiveCap tube (Zeiss). 10 fragments of either apical or basal sample type from different egg chambers were pooled for each replicate, with a total microdissected area of ~3000–4000 μm²/replicate. LCM samples were processed according to Chen et al.[88] to produce high-quality Illumina sequencing libraries. Samples were multiplexed and simultaneously sequenced in a single lane using the NextSeq500 system according to the manufacturer's instructions.

### RNA-seq analysis
Pre-processing of demultiplexed raw reads was performed on EMBL's instance of Galaxy platform. Raw reads were trimmed to remove low-quality bases, filtered from rRNA reads, and mapped against *D. melanogaster* Release 6 (dm6) reference genome. To control for RNA degradation that might have occurred during LCM, the normalized transcript coverage of the uniquely mapping reads was calculated with CollectRNAseqMetrics (part of Picard tools, http://broadinstitute.github.io/picard/). Uniquely mapped read counts were normalized with DESeq2[89]. Differential gene expression analysis was performed with DESeq2 by comparing the mean read counts of the Apical (4 replicates, A1-A4) and Basal (4 replicates, B1-B4) samples. Replicates A5 and B5 were excluded from further analysis due to their high degree of dissimilarity with replicates of the same sample type as shown by PCA and Euclidean distance analysis, probably due to a high degree of contamination from neighboring tissues. Statistical significance was set to an FDR-adjusted *p* value <0.1 (Benjamini-Hochberg correction for multiple testing). The R package ComplexHeatmap[90] was used to generate the heatmap in Fig. 2b.

### Identification of contaminant reads
Identification of contaminant RNAs was performed with R Studio. Among the RNAs that were significantly enriched in either the apical (log2FC > 0) or the basal (log2FC < 0) domain, were considered contaminants those RNAs displaying high absolute log2FoldChange (|log2FC|), indicating that they were probably originating from neighboring tissues. A threshold of log2FC > 3 and log2FC < −3 was arbitrarily set to identify putative apical and basal contaminants, respectively. The functional annotation of each contaminant candidate was retrieved on FlyBase[91] (release FB2020_6) and their read distribution among apical and basal replicates analyzed through Integrative Genomics Viewer (IGV)[92].

### Fly stocks and genetics
All fly stocks were maintained at 18 °C on standard fly food. For crosses, virgin females were mated with *w1118* males at 25 °C on cornmeal food supplemented with yeast. Female offspring of the desired genotype were incubated with *w1118* males on a yeast-supplemented medium for 24 h at 25 °C to stimulate the development of vitellogenic stage egg-chambers before ovary dissection.

The following stocks were obtained from the Bloomington Drosophila Stock Center (BDSC): *w1118* (wild-type; #3605), *DhcRNAi* (#36698), *eglRNAi* (#28969), *KhcRNAi* (#35409), *UAS-NLS-mCherry* (#38425), *osk-Gal4* (#44242), *VK33* (#9750). Other stocks used were: *HsFLP; arm > f + >Gal4; UAS-CD8-mCherry* and *tj-Gal4/CyO* (gifts of Juan Manuel Gomez Elliff), *Tm1eg1/TM3Sb,Ser* and *Tm1eg9/TM3Sb,Ser*[21], *GFP-Mago*[93], *Dhc64C-GFP*[68], *vasa-Gal4/TM3Sb* (gift of Jean Rene Huynh), *UAS-ΔC-Pym-GFP*[51], *UAS-Egl*[94], *eglWUSO/SM1* and *eglPR29/SM6A*[24]. Transgenic flies carrying *UAS-GFP, UAS-⁰BicD-GFP, UAS-⁽⁺¹⁾BicD-GFP*, and *UAS-⁽⁻¹⁾BicD-GFP* were generated in this study by phiC31 integrase-mediated recombination using the VK33 line, which carries an attP site on the third chromosome.

For the generation of *eglNULL* FC flies, *eglWUSO/CyO; osk-Gal4/TM3Ser* were crossed with *eglPR29/CyO; UAS-Egl/TM3Ser* to generate *eglWUSO/eglPR29; osk-Gal4/UAS-Egl*, expressing Egl only in the germline lineage to rescue the formation of rudimentary ovaries. *tj-Gal4* and *vasa-Gal4*

drivers were used to express UAS-containing transgenes in the whole follicular epithelium and in the germline, respectively.

To generate flies for FC mutant clone induction, male flies carrying a UAS-containing transgene were crossed with *hsFlp; arm > f + > Gal4; UAS-CD8-mCherry* virgins, and F1 females were subjected to heat-shock as described below.

To generate flies for induction of FC mutant clones in the experiment illustrated in Fig. 3, *HsFLP; arm > f + >Gal4/CyO; KhcRNAi/TM6B,Tb* flies were crossed with *+ ; UAS-NLS-mCherry/CyO; eglRNAi/TM3Ser*. F1 Female flies the desired genotypes [*eglRNAi/TM6B,Tb* for the *egl* RNAi condition; *KhcRNAi/TM3Ser* for the *Khc* RNAi condition; *eglRNAi/KhcRNAi* for the (*egl + Khc*) RNAi condition] were collected and subjected to heat-shock as described below.

### Generation of follicle cell clones
The UAS-Gal4 "flip-out" system was used to generate marked mutant clones in a wild-type background[95,96]. Freshly eclosed females resulting from each cross were collected and mated with *w1118* males for 24 h at 25 °C on food supplemented with yeast. Flies were heat-shocked for 1 h in a water bath heated at 37 °C. According to Gonzalez-Reyes & St Johnston[97], heat-shocked females were kept for 39 h at 25 °C with males on yeast before dissection, thus allowing follicle cells that induced the expression of the transgene at stage ~5 to develop into stage 10 follicle cells.

### Ex vivo pharmacological treatment
Young *w1118* female flies were incubated with males for 24 h at 25 °C on fly food supplemented with yeast. Ovaries were dissected in PBS and immediately incubated in Schneider's medium (Gibco) supplemented with 15% FBS (Gibco), 0.6X penicillin/streptomycin (Invitrogen), 200 μg/ml insulin (Sigma). For translation inhibitor treatment, either 200 μg/ml puromycin (Gibco) or 200 μg/ml cycloheximide (Sigma) or no compound (control) was added fresh to the medium and ovaries were incubated for 30 min at RT before fixation.

### Generation of BicD-GFP constructs and transgenic fly lines
AttB-pUASp-BicD-GFP-K10 or AttB-pUASp-GFP-K10 plasmids carrying a *w* + cassette, a TLS-deficient version of the K10 3'UTR, and attB sites for phiC31 integrase-mediated recombination into the VK33 line were generated as follows.

To generate plasmid vectors carrying the BicD-GFP gene cassettes (*⁰BicD-GFP, ⁽⁻¹⁾BicD-GFP, ⁽⁺¹⁾BicD-GFP*), BicD and GFP CDS were amplified by PCR and the two fragments were combined into AttB-pUASp-K10 vector by InFusion cloning (Clontech) according to the manufacturer's instructions. pBS-BicD (BicD-RA, FlyBase ID: FBpp0080555) plasmid (a kind gift from Jean-Baptiste Coutelis) was used as template to generate BicD CDS PCR amplicons. The Fw primer used to amplify BicD CDS was designed in order to include, in addition to a 20 nt-homology with AttB-pUASp-K10 vector, the *Drosophila* Kozak sequence[98] in frame with a linker sequence where frameshift mutations could be generated, and a region annealing to nt 4-29 of BicD CDS. To generate ⁰BicD-GFP construct, the 18-bp linker containing the ATG (5'- AT̲G̲ATCCT AGGCGCGCGG- 3') was inserted in frame with nt 4-2346 of BicD-RA. To generate ⁽⁺¹⁾BicD-GFP construct, a C was inserted at position 4 in the N-terminal 18-bp linker (5'- ATGC̲ATCCTAGGCGCGCGG- 3'). To generate ⁽⁻¹⁾BicD-GFP construct, a G was deleted at position 10 in the N-terminal 18-bp linker (5'- ATGATCCTA_G̲CGCGCGG- 3'). ⁰BicD-GFP, ⁽⁻¹⁾BicD-GFP, and ⁽⁺¹⁾BicD-GFP full insert sequences with the respective predicted translated ORF are listed in Supplementary Data 2.

To generate UAS-GFP construct, GFP ORF was amplified with a Fw primer containing KpnI restriction site upstream of GFP ATG and with a Rev primer containing NotI restriction site and the stop codon. The amplified fragment was gel purified, digested with KpnI and NotI and ligated into a AttB-pUASp-K10 vector digested with the same enzymes.

Each AttB-containing plasmid was purified and sequenced before injection into VK33 embryos carrying an attP site on the 3<sup>rd</sup> chromosome. Injected flies were crossed with *If/CyO; Sb/TM3Ser* individuals and transgenic F1 flies were identified by appearance of red eye color.

## Immunostaining

5–10 pairs of ovaries were dissected in PBS and immediately fixed in 2% PFA in PBSTX(0.1%) (PBS + 0.1% Triton-X100) on a Nutator for 20 min at RT, followed by two washes of 15 min each with PBSTX(0.1%) shaking at RT. Ovaries were then blocked in 1X casein/PBSTX(0.1%) (stock: 10X casein blocking buffer, Sigma) for 30 min and incubated with rabbit anti-Egl primary antibody[24] (kind gift from R. Lehmann) diluted 1:1000 in blocking buffer o/n at 4 °C. Alexa fluor 647 goat anti Rabbit (Jackson Immuno Research) secondary antibody was added at 1:750 dilution in blocking buffer for 2 h at RT. Samples were washed 3 × 10 min with 1X casein/PBSTX(0.1%), 1 × 10 min with PBSTX(0.1%) + 1:15,000 DAPI and kept o/n in 100 µl of 80% TDE/PBS before mounting on microscope slides.

## Single molecule in situ Fluorescence Hybridization (smFISH)

smFISH antisense oligonucleotides (listed in Supplementary Data 3) were designed and labelled with dye-conjugated ddUTPs according to the protocol described by Gáspár et al.[99] to generate oligonucleotides labelled at their 3' and with ATTO-633-NHS ester (ATTO-TEC). When dual-color smFISH experiments were performed, each probe set was labelled with either ATTO-633 or ATTO-565. The degree of labelling (DOL, % of labelled oligos) and concentration of the labelled probe sets was measured according to the published algorithm.

Dissected ovaries were immediately fixed in 2% PFA/PBSTX(0.1%) gently shaking for 20 min at RT. In case of ex vivo ovary incubation, dissected ovaries were incubated in Schneider's medium supplemented with the respective pharmacological treatment before proceeding with fixation, as described above. Fixed ovaries were rinsed and washed twice with PBSTX(0.1%) for 10 min before dehydrating them by replacing PBSTX(0.1%) with increasing concentrations of ethanol/PBSTX(0.1%). Fixed and dehydrated ovaries were kept in 100% ethanol at −20 °C for up to 10 days until the day of the experiment.

An optimized version of the smFISH protocol described in Hampoelz et al.[100] was followed with minor modifications. All steps were performed at RT unless specified otherwise. Dehydrated ovaries were first rinsed with PBSTX(0.1%), followed by 2 × 15 min washes with PBSTX(0.1%), and incubated in Pre-hybridization Buffer (2 x SSC, 10% deionized formamide, 0.1% Tween-20) gently shaking for 30 min. The Pre-hybridization Buffer was replaced with 250 µl of Hybridization Buffer (2 x SSC, 10% deionized formamide, 0.1% Tween-20, 2 mM vanadyl ribonucleoside complex (New England Biolabs), 100 µg/mL salmon sperm DNA (Invitrogen), 10% dextran sulfate, 20 µg/mL BSA) pre-warmed at 37 °C in which smFISH probes were added to a final concentration of 1 nM/probe. Ovaries were kept hybridizing in the dark for 16-17 h on a heat block set at 37 °C shaking at 1000 rpm. To remove the excess probes, ovaries where washed 3 × 10 min at 37 °C with Washing Buffer (2 x SSC, 10% deionized formamide, 0.1% Tween-20). 1:15,000 DAPI was added to the second wash. Finally, samples were rinsed 4x in PBST(0.1%) (PBS + 0.1% Tween20) and kept in in 100 µl of 80% TDE/PBS for at least 1 h before mounting on microscope slides.

Z-stacks of images were acquired on a Leica TCS SP8 confocal microscope with 405 nm, 488 nm, 552 nm and 640 nm fixed excitation laser lines using a 63 × 1.3 NA glycerol immersion objective. A suitable range for spectral detection was carefully chosen for each channel to avoid cross-talk of fluorescence emission. Images were automatically restored by deconvolution with the Lightning module.

## Quantification of smFISH signal

To quantify smFISH fluorescence of localizing RNAs, average Z-projections of deconvolved confocal image stacks were analyzed

with Fiji[101]. In mosaic FE, for each wild-type (wt, unmarked) and mutant (mCherry-marked) group of cells within the same Z-stack, a region of interest (ROI) was drawn encompassing the apical and the basal cytoplasm of 5-10 adjacent follicle cells (with the exclusion of nuclei); in addition, a ROI was drawn in an area of the image were no signal was present (background, bg). The mean fluorescence intensity (m.f.i.) was measured for each ROI.

The degree of apicality (DoA) of a given RNA in each cell type (*t*) (wild-type or mutant) and each experimental condition *c*, was measured as follows:

$$DoA_{t,c} = \left( \frac{Apical\ m.f.i._t - bg\ m.f.i}{Basal\ m.f.i._t - bg\ m.f.i} \right)_c \quad (1)$$

To quantitatively analyze changes in RNA localization in each experimental condition *c*, the DoA measured in mutant (KD) cells was divided by the DoA measured in neighboring wild-type (wt) cells within the same Z-stack:

$$KD/wtDoA_c = \left( \frac{DoA_{KD}}{DoA_{wt}} \right)_c \quad (2)$$

Only in *Tm1^NULL* condition, due to the impossibility to obtain a mosaic tissue, the DoA of a given RNA in each cell type (*t*) (wild-type or *Tm1^NULL*), was measured as follows:

$$DoA_t = \left( \frac{Apical\ m.f.i - bg\ m.f.i}{Basal\ m.f.i - bg\ m.f.i} \right)_t \quad (3)$$

To calculate the change in DoA, the DoA measured in single *Tm1^NULL* egg chambers was divided by the average DoA measured in *n* wild-type egg chambers (wt):

$$Tm1NULL/wt\ DoA = \frac{DoA_{Tm1NULL}}{\frac{1}{n}\sum DoA_{wt}} \quad (4)$$

## Bioinformatic analysis of apical and basal RNAs

To analyze EJC enrichment on apical and basal RNAs, we annotated differentially EJC-enriched genes from Obrdlik et al.[52] as apical or basal according to our RNA-seq analysis results, excluding putative oocyte and muscle contaminants, and plotted their log2FoldChange(EJC vs. RBP). The p-value was estimated by double-sided Wilcoxon rank sum test.

To calculate translation efficiency of apical and basal RNAs, ribosome profiling (RFP) TPM values as well as RNA-seq TPM values were extracted from published ribo-seq datasets of 0-2 h embryos[55] and control RNAi ovaries[54]. Translation efficiency was calculated as RFP[TPM]/RNA-seq [TPM] for apical and basal RNAs identified by our study. The p-value was estimated by double-sided Wilcoxon rank sum test.

## Statistics and reproducibility

Immunostaining and/or smFISH experiments were performed on ovaries from at least 3 flies (hundreds of egg chambers) and repeated at least twice at different times.

In Figs. 2d, 4d, and 6c, KD/wt DoA values of each RNA and experimental condition was measured across at least 3 different Z-projections, and the average value obtained was compared to the null hypothesis $H_0$: KD/wt(DoA) = 1, corresponding to no change in RNA localization bias following KD treatment [DoA(KD)=DoA(wt)]. One-sample two-sided Student's *t*-test was used to compare means to a reference value of mu = 1 in each experimental condition. In Supplementary Figure 2c-d, we calculated the total smFISH signal intensity as the sum of apical and basal signals in mutant and wild-type cells,

computed the KD/wt total signal intensity and tested against a null hypothesis of $H_O$: KD/wt(total signal intensity)=1, corresponding to no change in total signal intensity in KD cells. One-sample one-sided Student's *t*-test (alternative="less") was used to compare means to a reference value of mu = 1 in each experimental condition. In Supplementary Fig. 4d, GFP-Mago fluorescence intensity was measured in apical and basal domains. The A/B ratio of GFP intensity was tested against a null hypothesis of $H_O$: A/B=1, corresponding to an evenly distributed signal long the A-B axis. One-sample two-sided Student's *t*-test was used to compare mean GFP-Mago fluorescence to a reference value of mu = 1.

In Fig. 3b, independent two-sided Student's *t*-test was used to compare mean DoA(wt) and DoA(RNAi) values in each condition. In Fig. 3c, mean KD/wt DoA values across conditions were compared by one-way ANOVA followed by Tukey's post-hoc tests.

Fluorescence intensity along lines was measured with Fiji on average Z-projections of confocal images and plotted with R Studio. Intensity values from each channel were normalized to 0-1 range.

*BicD* or *GFP* mean fluorescence intensity along the A-B axis of the epithelium was measured in groups of 5–10 adjacent follicle cells as line plots. At least 3 line plots were generated for each RNA measured in each condition. The value corresponding to 50% of the cumulative area under the curve (a.u.c.) of each plot was considered as the variation of the respective RNA localization along the A-B axis of the epithelium. Welch two sample *t*-test was used to compare mean values of the 50% of the a.u.c with respect to untreated controls (pharmacological experiments) or In-frame BicD-GFP (Frameshift vs. In-frame variation).

### Quantification of *BicD*, *hook*, and *Bsg25D* signal following translation inhibition

smFISH signal in Control, CHX, and Puro conditions was quantified using Fiji. Due to low signal-to-noise ratio of *Bsg25D* and, to a lesser extent, *hook*, smFISH signal threshold was first manually adjusted. Then, the integrated density (IntDen) of background-subtracted signal was measured in the apical-cortical, subapical, and basal domains of adjacent FCs. To minimize sample-to-sample variability due to differences in threshold adjustment, the percentage of signal present in each subdomain within single egg chambers was measured, and average percentages were plotted for each RNA in each condition. Statistical testing was performed by independent two-sided Student's *t*-test ($n = 4$ replicates per RNA analyzed).

### Reporting summary

Further information on research design is available in the Nature Research Reporting Summary linked to this article.

## Data availability

The data supporting the findings of this study are available within the article and its supplementary information files. Raw microscopy images are available upon request from the corresponding author. The D. melanogaster genome release 6 (dm6) data used in this study are available in the NCBI GenBank assembly database under accession code GCA_000001215.4. The raw RNA-seq data generated in this study have been deposited in the ArrayExpress database under accession code E-MTAB-9127. The processed RNA-seq data are provided in the Supplementary Files (Supplementary Data 1). Source data are provided with this paper.

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

## Acknowledgements

We thank the Bloomington Drosophila Stock Center (BDSC), J. M. Gomez Elliff, A. Debec, I. Gaspar, S. Bullock, J. R. Huynh and R. Lehmann for providing fly lines and reagents. We are grateful to I. Gaspar and K. Zarnack for helpful discussions and training. We are grateful to A. Obrdlik and Lin Gen for discussion of the IpaRT dataset. We thank the EMBL GeneCore Facility (D. Pavlinic, V. Benes), Advanced Light Microscopy Facility (S. Terjung), Drosophila Injection Service (A. Reversi), Centre for Bioimage Analysis (C. Tischer), Centre for Statistical Data Analysis (B. Klaus), Genome Biology Computational Support (C. Girardot), and W. Huber for their support. We are grateful to K. Zarnack, P. Ronchi, L. Russo, A. Reversi and members of the Ephrussi lab for critically reading the manuscript. This work was supported by EMBL. L.C. was supported by DFG-FOR 2333 grants EP 37/2-1 and EP 37/4-1 from the Deutsche Forschungsgemeinschaft (Germany) to A.E.

## Author contributions

Conceptualization: L.C. and A.E.; Investigation: L.C.; Data analysis: L.C; Writing – original draft: L.C.; Writing – review & editing: L.C. and A.E; Supervision: A.E.; Funding acquisition: A.E.

## Funding

## Competing interests

The authors declare no competing interests.
