## [Peer Review File · Nature Communications]

Subcellular spatial transcriptomics identifies three mechanistically different classes of localizing RNAsREVIEWER COMMENTS

Reviewer #1 (Remarks to the Author):

In this manuscript presented by Cassella and Ephrussi, the authors set out to study the targeting mechanisms of mRNAs displaying differential localization along the apico-basal axis of follicular epithelial (FE) cells in *Drosophila* ovaries. By performing laser-capture microdissection and RNA-seq, they first identify collections of mRNAs that are differentially localized to the apical and basolateral compartments of FE cells, followed by FISH-based validation assays. This work was then complemented by genetic studies to begin characterizing the determinants of mRNA localization to the apical and basal compartments of FE cells. Specifically, the authors found that localization of basal mRNAs is disrupted following genetic perturbation of Kinesin-1, aTm1 or of the EJC pathway component PYM. By contrast, the localization of several apical mRNA candidates is depended on Dynein, BicD and Egl, while the apical targeting of several transcripts encoding Dynein-interacting adaptors (e.g. BicD, Bsg25D) is dependent on Dynein and translation. This manuscript presents a novel dataset of differential localization activity in FE cells, while also extending previous insights into the mechanisms driving their localization. The experiments are convincingly and well conducted. The paper is also very clearly constructed and well written.

Main Criticisms/Questions:

- The requirement of the EJC component PYM for basal mRNA localization is interesting, but it also raises questions of how a regulatory complex that binds most spliced mRNAs may have such selective impacts on transcript localization. Moreover, considering the recent study by Kwon et al. (PMID: 27422905), which revealed a role of the EJC in mRNA localization to centrosomes and basal bodies in mammalian cells, one might expect a functional contribution for the EJC in apical mRNA targeting as well (i.e. the compartment where centrosomes are found). Would disruption of other EJC components lead to similar phenotypes as PYM loss-of-function? Is there any evidence that PYM or other EJC components are preferentially associated with the basal class of mRNAs, e.g. from previous binding studies such as PMID: 31365866 ? Could the authors strengthen the study by adding immunofluorescence data for specific EJC components to assess where they are localized in FE cells? These points should at least be addressed adequately in the Discussion section.
- To study more globally the impact of genetic perturbations on mRNA localization in FE cells, did the authors consider performing laser-capture microdissection on specific mutant tissue specimens or on specimens treated with translational inhibitors? Such data could help support some of the primary conclusions of the study.
- The following phrase on line 221-222 is confusing: 'We noticed that, as in FE, BicD RNA localization to the posterior of the oocyte (stages 9-10) was impaired in Puro-treated ovaries, whereas CHX had no effect (Fig.4C,D)'. Wasn't the finding from Figure S6A that BicD RNA accumulates at the anterior corners in stage 9-10 oocytes, not at the posterior end. Following Puro treatment, it seems that the localization of BicD mRNA remains at anterior corners, but with a more diffuse pattern.
- It seems that data from Fig.S3 should be included in one of the main figures.
- In Fig.S6C, it would be important to include a side-by-side comparison of the frameshift transgene with the control 'in frame' construct, which would help confirm that the 'in frame' RNA localizes comparably to the endogenous.

Minor Points :

- The authors should clearly indicate which frameshift variant is being studied in each relevant figure

panel.

- With regards to citations, the authors should add the following citation (PMID: 32160541), concerning the co-targeting of Cen and Ik2 mRNAs to centrosomes in fly embryos, to the phrase on lines 19-21 mentioning hitch-hiking mechanisms of RNA localization. Also, previous work has mapped localization determinants of the Bsg25D transcript could be relevant to mention (PMID: 27422905), since this study showed that the coding region of Bsg25D is sufficient for targeting microtubules in fly embryos.

- The sentence on line 118-119 should be edited for precision, since the affirmation is true only for the RNAs tested by RNA FISH.

Reviewer #2 (Remarks to the Author):

In this manuscript, Cassella and Ephrussi uncover mechanisms that underlie RNA localization across the apicobasal axis of follicular epithelia in *Drosophila* embryos. Overall, this manuscript details exciting new findings, and the data presented and conclusions drawn are generally sound. I have only a few comments to make that may improve the manuscript.

MAJOR COMMENTS

1. The authors make repeated use of mosaic RNAi lines. These are incredibly powerful and are well-presented. The authors ask how the localization of endogenous and reporter transcripts changes in RNAi cells compared to neighboring unaffected cells. The metric reported for these experiments is always a proportional metric (e.g. DoA) in which signal from apical and basal regions is compared. It could also be useful to know whether the overall number of transcript molecules (or at least amount of smFISH fluorescent signal) is different between the RNAi and unaffected cells. Put another way, is the overall expression/stability of the RNA changing in addition to its localization, or is it the same amount of RNA just ending up at different subcellular locations?

This may be especially relevant to the frameshift constructs. Are these constructs NMD targets?

2. The authors found that RNA encoding the dynein adapter BicD is apically localized in FE cells. They then ask if RNA encoding other dynein adapters is similarly apically localized, and whether or not this requires active translation (Fig 5E-G). For one of the RNAs investigated, hook, there is reasonable evidence that its apical localization requires ribosome association with transcripts as it is sensitive to puromycin but not cycloheximide. For the other transcript interrogated, though, (Bsg25D), the data are much, much weaker. There is no significant difference in the localization of Bsg25D transcripts in control, cycloheximide, and puromycin treatments. Yet the text still says "the apical localization of both adaptor RNAs showed sensitivity to Puro but not CHX." I do not agree with this statement as (1) there is no significant difference between the two treatments, and (2) there is no difference at all between the control and puromycin conditions. This statement should be toned down or removed.

3. If the authors wish to argue more forcefully for the translation-dependent apical localization of Bsg25D RNA, perhaps they could employ the frameshifted reporter approach they used for BicD in Figure 4. This would also remove the technical hurdle of dealing with the low expression level of endogenous Bsg25D.

MINOR COMMENTS

1. For BicD RNA, the authors say that "the first peptide emerging from the ribosome is the dynein-

binding domain." They use this as possible support for a model in which the RNA's apical localization is cotranslational. Is the dynein-binding domain in the other tested dynein adapters also N-terminal?

2. The schematics alongside some supplementary figures (e.g. Fig S2A, S4A) are very helpful for understanding the experiment. Perhaps it could be considered to include them in their analogous places in the main figures.

3. Similarly, the anterior to posterior line across the bottom of the plots in figure S6B is very helpful. Perhaps an analogous addition could be made to the plots in 4E and 5F (in this case apical to basal) to make what is being shown more clear.

Reviewer #3 (Remarks to the Author):

Cassella and Ephrussi present a considered study that provides a significant advance in our understanding of mechanisms of RNA localization in a rarely-studied system, the follicular epithelium (FE) of *Drosophila* ovaries. The authors first identified a novel pool of apically and basally localized RNAs in the FE using laser-capture microdissection followed by RNAseq analysis of the microdissected tissue. They conducted a number of QC steps such as filtering out likely contaminants and instituting a fairly rigorous threshold of statistical significance, along with validation of some identified RNAs using smFISH. They then tested how knockdown or removal of specific components of transport various machineries affected apical or basal localization of RNAs, and identified three main pathways: 1) an "always-on" dynein- and egalitarian-dependent apical pathway, 2) a kinesin-1-dependent basal pathway that overrides the first pathway for basally localizing RNAs, and 3) a dynein-dependent, translation-dependent, egalitarian-independent pathway that mediates transport of RNAs encoding dynein adaptor proteins such as BicD. Missing from this work, however, is consideration of the functional consequences of these localization pathways. Inclusion of such data should be required for publication in *Nature Communications*.

Other Concerns:

1. The images of the microdissected tissue (lower right in Fig. 1A) are unnecessary. This reviewer recommends moving them to a supplementary figure or deleting them, as they don't illustrate any particular scientific point.

2. A supplementary figure should be included to illustrate how the ROIs are drawn for calculation of Degree of Apicality (DoA), as it's unclear whether the ROIs are essentially dividing the cell in half (minus the nuclei) or if they are drawn closer to the schematic shown in Fig. S1A, closer to the edges of the cells. Given the patterns of localization, those two different methods would likely lead to different calculations, which would impact reproducibility.

3. As depicted in Fig. 1C, Dlic and Imp do not appear to be strongly apical while AdiopR and Rtnl1 do not appear to be strongly basal.

4. It's somewhat concerning given the pattern of localization in Fig. 4J that Dhc64C RNA isn't found in the list of apical RNAs. This should be addressed in the text (perhaps the statistics were not sufficiently consistent?).

5. The elucidation of the BicD RNA localization pathway in oocytes (Supplementary Figs. 5 and 6), while interesting, strays from the focus of the manuscript.

Minor Concerns and/or Edits (with line numbers)

Fig. 1A: Scale bars are needed for the images.

Fig. 1A: The use of yellow for the dashed lines around the images of the FE makes it nearly impossible to see. A different color should be used

Line 22: Change "consists in" to "consists of".

Line 24: Because encoded implies translation, recommend changing "encoded" to "contains".

Line 71: Change "consists in" to "consists of".

Line 76: Change "aimed at identifying" to "aimed to identify".

Line 99: "Functionally equivalent" seems overstated. Recommend changing to "similar".

Line 123: Recommend adding "in the absence of kinesin-1".

Line 247: The term "adaptor RNAs" makes it sound like the RNA itself is the adaptor. Recommend changing to "adaptor-encoding RNAs".

Line 248: Delete "with the exception of". Two examples (Nuf and Milton) are not affected and two examples (Bsg25D and hook) are affected. So, Nuf and Milton cannot be described as an exception to a rule.

Line 765: Change "showed" to "shown".

REVIEWER COMMENTS

Reviewer #1 (Remarks to the Author):

In this manuscript presented by Cassella and Ephrussi, the authors set out to study the targeting mechanisms of mRNAs displaying differential localization along the apico-basal axis of follicular epithelial (FE) cells in *Drosophila* ovaries. By performing laser-capture microdissection and RNA-seq, they first identify collections of mRNAs that are differentially localized to the apical and basolateral compartments of FE cells, followed by FISH-based validation assays. This work was then complemented by genetic studies to begin characterizing the determinants of mRNA localization to the apical and basal compartments of FE cells. Specifically, the authors found that localization of basal mRNAs is disrupted following genetic perturbation of Kinesin-1, aTm1 or of the EJC pathway component PYM. By contrast, the localization of several apical mRNA candidates is depended on Dynein, BicD and Egl, while the apical targeting of several transcripts encoding Dynein-interacting adaptors (e.g. BicD, Bsg25D) is dependent on Dynein and translation. This manuscript presents a novel dataset of differential localization activity in FE cells, while also extending previous insights into the mechanisms driving their localization. The experiments are convincingly and well conducted. The paper is also very clearly constructed and well written.

We thank the reviewer for appreciating the quality of our work and the further insight it contributes into the mechanisms driving RNA localization in an epithelial tissue in vivo.

Main Criticisms/Questions:

- The requirement of the EJC component PYM for basal mRNA localization is interesting, but it also raises questions of how a regulatory complex that binds most spliced mRNAs may have such selective impacts on transcript localization. Moreover, considering the recent study by Kwon et al. (PMID: 27422905), which revealed a role of the EJC in mRNA localization to centrosomes and basal bodies in mammalian cells, one might expect a functional contribution for the EJC in apical mRNA targeting as well (i.e. the compartment where centrosomes are found).

According to our data (Figure 2C, Supplementary Figure 4A of the revised documents, former Figure 2C and Supplementary Figure 2D) the EJC is not involved in apical RNA localization. Rather, like Khc and aTm1, the EJC contributes to the localization of basal RNAs to MT plus ends. These results agree with previous findings highlighting that the EJC participates with Khc and aTm1 in the localization of *oskar* RNA to the posterior pole of the oocyte, where MT plus ends are enriched.

Interestingly, Kwon et al. (PMID: 27422905) reported that one RNA (*NIN*) is localized to centrosomes in an EJC-dependent fashion in mono-ciliated cells. Since *NIN* protein is a core component of the centrosome (PMID: 15784680), and no centrosomal localization of the EJC was observed by Kwon et al. in multi-ciliated cells, the EJC-dependent localization of *NIN* RNA to centrosomes appears to be a characteristic of quiescent mono-ciliated cells, such as mNSC and RPE1. In *Drosophila*, during mid-oogenesis, the columnar follicle cells (~stage 9-10) lose apical centrioles (PMID: 104872) and acquire non-centrosomal MTOCs as they mature from cuboidal to secretory columnar epithelium (PMID: 18304845; PMID: 27404359), similar to mid-stage oocytes (PMID: 27404359). Therefore, despite maintaining a conserved role in RNA localization, the mammalian EJC might be specifically required for the localization of *NIN* RNA at basal bodies of mono-ciliated cells. In contrast, the *Drosophila* EJC appears to contribute to the MT plus end-directed localization of several RNAs through a centrosome-independent mechanism in both the somatic follicular epithelium (basal RNAs) and the germline (*oskar* RNA).

We now comment on this in the Discussion section (lines **316-325** of the revised Main text): “Interestingly, Kwon et al.²³ found that the EJC specifically localizes to the basal body of the primary cilium in mono-ciliated cells, where it controls the centrosomal localization of *NIN* RNA towards MT minus ends. In contrast, our data suggest that in columnar follicle cells, which are characterized by non-centrosomal MTOCs⁶⁴⁻⁶⁶, the EJC may play a role in MT plus end-directed (basal) RNA transport by acting synergistically with kinesin-1 and aTm1. Strikingly, the localization of *osk* RNA to the posterior pole of the oocyte also relies on the presence of MTs generated from non-centrosomal MTOCs⁶⁵. Therefore, the mammalian EJC might have acquired a specific role in the localization of *NIN* RNA at basal bodies of mono-ciliated cells, while the *Drosophila* EJC appears to

contribute to the MT plus end-directed localization of several RNAs through a centrosome-independent mechanism both in the somatic follicular epithelium (basal RNAs) and in the germline (*osk* RNA).”

Would disruption of other EJC components lead to similar phenotypes as PYM loss-of-function?

We have not tested directly if disruption of individual EJC components leads to similar phenotypes as over-expression of PYM (gain of function), but we believe that this would be the case, for the following reasons.

Homozygous mutation of EJC components in the fly has been shown to cause lethality, and mosaic mutant egg chambers display aberrant cellular morphology and cell death (PMID: 14973490) and abnormal microtubule organization (PMID: 11691839). To circumvent these issues and address the role of the EJC in RNA localization *in vivo*, we previously implemented an effective strategy (PMID: 24967911) that relies on EJC disassembly and displacement by overexpression of Pym protein (PMID: 19410547). We also previously showed that Pym lacking its C-terminal domain (deltaC-Pym) is most effective in disrupting the EJC (PMID: 24967911). Overexpression of deltaC-Pym causes EJC disassembly and impairs *oskar* RNA localization to the posterior pole (PMID: 24967911), in a manner that phenocopies mutation of the individual EJC components (Y14: PMID: 11696323, PMID: 11691839; Mago nashi: PMID: 8026338; eIF4AIII: PMID: 14973490; Barentsz: PMID: 11481346). Furthermore, as no cellular phenotype has been associated with Pym overexpression, displacing the EJC by Pym overexpression avoids the emergence of confounding phenotypes that might affect RNA localization indirectly. Therefore, we would expect that also in the FC, Pym over-expression would phenocopy disruption of the core EJC components.

Is there any evidence that PYM or other EJC components are preferentially associated with the basal class of mRNAs, e.g. from previous binding studies such as PMID: 31365866 ?

We thank the referee for this very interesting question, which led us to perform further in-depth analysis of EJC-dependent RNA localization, which we describe in a new paragraph in the Results section (lines **121-141** of the revised Main text), in **Supplementary Figure 4**, and in Materials and Methods (line **454**, lines **577-586**, lines **598-602**).

We computationally investigated whether the EJC preferentially associates with basal RNAs by comparing EJC-enriched and EJC-depleted RNAs identified by Obrdlik et al. (2019) (PMID: 31365866) with apical and basal RNAs found in our study. We annotated apical and basal RNAs resulting from our RNA-seq as EJC-enriched or EJC-depleted RNAs according to data retrieved from PMID: 31365866. We retrieved this information for 288 apical RNAs and 214 basal RNAs and plotted their EJC vs. RBP fold change as a proxy for EJC enrichment. The result of this analysis shows that, in fact, basal RNAs are on average EJC-depleted [$\log_2FC(\text{EJC vs. RBP}) = -0.51$]; in contrast, apical RNAs are on average EJC-enriched [$\log_2FC(\text{EJC vs. RBP}) = 0.37$] (**Supplementary Figure 4B**). This result is consistent with a slight apical enrichment of cytoplasmic GFP-Mago, as seen by GFP-Mago fluorescence intensity (**Supplementary Figure 4C,D**, see below).

Due to the basal-specific effect of EJC displacement, these results were somewhat surprising, as we expected basal RNAs to be on average EJC-enriched. However, since the EJC is displaced from actively translating RNAs (PMID: 12121612), RNAs that are on average more translated should have less EJC bound at exon junctions. Therefore, we hypothesized that basal RNAs appear on average EJC-depleted due to a higher translation rate as compared to apical RNAs. To test this hypothesis, we compared the translational status (measured as translational efficiency) of apical and basal RNAs using data from two independent *Drosophila* ribo-seq published datasets (wild-type ovaries, PMID: 30115809; 0-2h embryos, PMID: 31755866). In both embryos and ovaries, basal RNAs were significantly more translated than apical RNAs (**Supplementary Figure 4E**). This finding suggests that (1) basal RNAs are on average EJC-depleted due to their status of active translation, and (2) the discrepancy observed in the EJC enrichment between apical and basal RNAs depends on their differential translational efficiency. These differences in translational status did not interfere with the efficiency of EJC displacement by deltaC-Pym expression, as Dmel Pym dismantles EJCs in a translation-independent manner (PMID: 24967911). Therefore, the degree of EJC enrichment on classes of RNAs that display different translational efficiencies cannot be taken into consideration as a predictive factor to assess whether an RNA depends on the EJC for its localization.

The finding that basal RNAs are on average more efficiently translated than apical RNAs supports the leading hypothesis in the field, whereby the dynein/BicD/Egl transport machinery is part of a larger RNP complex that

ensures tight translational control of the transported RNA (PMID: 22666086). Therefore, RNAs transported apically in the FE by the dynein/BicD/Egl transport complex might be kept in a translationally silent state until they reach their final destination.

We now mention this in the Discussion section (lines **349-353**): “*The BicD/Egl machinery has been hypothesized to be part of a larger RNP complex that ensures a tight translational control of the transported RNA*²³. Our finding that basal RNAs are on average more translated than apical RNAs suggests that RNAs transported apically in the FE by the dynein/BicD/Egl transport complex might indeed be kept in a translationally silent state until they have reached their final destination”.

Could the authors strengthen the study by adding immunofluorescence data for specific EJC components to assess where they are localized in FE cells? These points should at least be addressed adequately in the Discussion section.

To investigate if the EJC shows a localized expression, we used a transgenic fly line expressing GFP-Mago from the *mago* promoter (PMID: 9272960); results are shown in **Supplementary Figure 4C-D**. GFP-Mago displays a predominant nuclear localization. In the cytoplasm, GFP-Mago is diffuse and present at lower levels throughout the apical and basal domains, consistent with its presence on most RNAs. As shown by the line profile in **Supplementary Figure 4C**, the intensity of GFP-Mago is non-homogeneous along the A-B axis, with a slight enrichment in the apical domain compared to the basal domain. The mean A/B ratio of GFP-Mago intensity is significantly higher than 1 (**Supplementary Figure 4D**), namely the condition of equal signal distribution, showing that GFP-Mago is enriched in the apical domain. These results agree with data presented above (**Supplementary Figure 4B**), indicating that cytoplasmic EJC enriches on apically-localizing RNAs, which display a lower translation efficiency than basal RNAs (**Supplementary Figure 4E**).

We have added the following sentence in the Discussion section: “*According to our analysis, deposition of the EJC is necessary but not sufficient for determining RNA localization, as we found that the EJC is deposited on both apically- and basally-directed RNAs.*” (lines **314-315** of the revised Main Text).

- To study more globally the impact of genetic perturbations on mRNA localization in FE cells, did the authors consider performing laser-capture microdissection on specific mutant tissue specimens or on specimens treated with translational inhibitors? Such data could help support some of the primary conclusions of the study.

The experiments the Reviewer is suggesting would give important insight regarding RNAs whose localization is affected by mutations in components of RNA transport complexes or displaying co-translational localization. It would be especially interesting to perform LCM and RNA-seq on *egl* RNAi egg chambers and Puro-treated egg chambers, for the transcriptome-wide identification of RNAs that are localized apically through non-canonical, co-translational transport. However, several technical limitations linked to laser-capture microdissection limit our ability to perform these experiments with the required resolution.

Our knock-down experiments were performed on FC clones to avoid the emergence of morphological defects and the disruption of the entire FE. To this end, we followed a heat-shock protocol that induced the expression of the shRNA at stage 5 for mutant cells visualized at stage 10 (PMID: 9655806). Laser capture microdissection on mutant FE would require to knock down the expression of each gene of interest in the whole epithelium by using classical mutants or RNAi, causing an exacerbation of defects such as nuclear mispositioning and disruption of cellular morphology upon lack of Dhc, *egl*, or Khc (PMID: 34160561; PMID: 27017624; PMID: 12225672; this study). In our experimental set-up, nuclear alignment is a pre-requisite for the collection of good quality intact tissue and allowed us to collect apical and basal subcellular fragments encompassing at least 5-10 adjacent cells. Nuclear mispositioning would restrict the area that can be cut for each fragment, dramatically affecting micro-dissected tissue quality (due to an increase in the surface damaged by the laser) and decreasing the efficiency of fragment collection.

Prolonged puromycin and cycloheximide treatments cause severe disruption of tissue morphology (data not shown). We therefore chose to perform a reduced treatment with translational inhibitors (30 min incubation in S2 medium, see Materials and Methods) to maintain an ovarian tissue morphology similar to that of untreated controls (S2 medium only). With such a treatment we could appreciate small but significant changes in RNA localization of the three dynein adaptor-encoding RNAs (*BicD*, *Bsg25D*, *hook*), especially between the apical-

cortical and subapical domain of follicle cells (see below). Due to fragment size constraints, laser-capture microdissection does not allow us to separate sub-domains of FE cells such as the apical-cortical and subapical fragments. As a result, comparison of the apical and basal transcriptome of 30 min inhibitor-treated egg chambers would fail to capture the changes in RNA localization we observed with microscopy.

-The following phrase on line 221-222 is confusing: 'We noticed that, as in FE, BicD RNA localization to the posterior of the oocyte (stages 9-10) was impaired in Puro-treated ovaries, whereas CHX had no effect (Fig.4C,D)'. Wasn't the finding from Figure S6A that BicD RNA accumulates at the anterior corners in stage 9-10 oocytes, not at the posterior end. Following Puro treatment, it seems that the localization of BicD mRNA remains at anterior corners, but with a more diffuse pattern.

We thank the referee for catching this error! We have corrected the sentence, which now reads "We noticed that, as in the FE, BicD RNA localization to the **anterior** of the oocyte (stages 9-10) was impaired in Puro-treated ovaries, whereas CHX had no effect" (lines 257-258 of the revised Main text).

- It seems that data from Fig.S3 should be included in one of the main figures.

We agree and in the revised manuscript have turned Supplementary Figure 3 into **Figure 3**.

- In Fig.S6C, it would be important to include a side-by-side comparison of the frameshift transgene with the control 'in frame' construct, which would help confirm that the 'in frame' RNA localizes comparably to the endogenous.

We have added the information comparing the frameshift with the in-frame construct, but also compacted the text as Reviewer#3 suggested (lines 254-268 of revised Main text). The new panel is shown in **Supplementary Figure 7C** (former Supplementary Figure 6C). Similar to endogenous *BicD*, in-frame *BicD-GFP* localizes in the oocyte at MT minus ends throughout oogenesis (data shown for stage 5 egg chambers), although its localization appears more biased towards the MT minus ends than endogenous *BicD*. We hypothesize that this higher accumulation at MT minus ends depends on an increased translational efficiency of *BicD-GFP* constructs that do not carry endogenous *BicD* UTRs (see Materials and Methods), which might be necessary for Me31B-mediated translational repression in the nurse cells (PMID: 11546740). As a result, as soon as in-frame *BicD-GFP* RNA is exported from nurse cell nuclei, it is co-translationally transported to the oocyte causing the higher accumulation of the transgenic in-frame RNA compared to the endogenous, translationally repressed *BicD* RNA. These data support our hypothesis of a nurse cell-to-oocyte translation-independent localization of *BicD* RNA, followed by a translation-dependent localization of the RNA at MT minus ends upon oocyte entry.

Minor Points :

- The authors should clearly indicate which frameshift variant is being studied in each relevant figure panel.

The frameshift construct used in **Figure 5G** (former Figure 4G) refers to ⁽⁻¹⁾*BicD-GFP* transgenic construct, although the same mislocalization phenotype was observed for ⁽⁺¹⁾*BicD-GFP*, which we did not show as a separate figure panel due to space constraints; however, we have added this information in the **Figure 5** legend: "Expression of different BicD-GFP constructs ("In-frame": ⁰*BicD-GFP*; "Frameshift": ⁽⁻¹⁾*BicD-GFP*) in FC clones and analysis of transgenic RNA distribution by smFISH with antisense GFP probes. Clones expressing each BicD-GFP construct are marked by CD8-mCherry (magenta). ⁽⁺¹⁾*BicD-GFP* frameshift RNA was mislocalized similarly to ⁽⁻¹⁾*BicD-GFP* (data not shown)" (lines 973-977).

In **Supplementary Figure 7C** (former Supplementary Figure 6C) of the revised manuscript, all three transgenic *BicD-GFP* constructs analyzed (⁽⁻¹⁾*BicD-GFP*: *Fshift(-1)*; ⁽⁺¹⁾*BicD-GFP*: *Fshift(+1)*; ⁰*BicD-GFP*: *In-frame*) are now shown.

- With regards to citations, the authors should add the following citation (PMID: 32160541), concerning the co-targeting of Cen and Ik2 mRNAs to centrosomes in fly embryos, to the phrase on lines 19-21 mentioning hitch-hiking mechanisms of RNA localization.

We have added the citation relative to PMID: 32160541 (ref. n. **15**) at line **21** of the revised Main text. We have also added two more citations as examples of RNAs hitch-hiking on other RNAs for their transport (PMID: 9130719; PMID: 22028360)

Also, previous work has mapped localization determinants of the *Bsg25D* transcript could be relevant to mention (PMID: 27422905), since this study showed that the coding region of *Bsg25D* is sufficient for targeting microtubules in fly embryos.

We thank the reviewer for this suggestion. As the reviewer correctly notes, the fact that *Bsg25D* CDS is sufficient for targeting *Bsg25D* protein to MT minus ends strengthens our finding of a translation-dependent mechanism of *Bsg25D* RNA localization. Accordingly, in the revised Discussion section we have added the sentence (lines **359-360**): “Moreover, both *Bsg25D*⁷⁴ and *BicD* (this study) RNA constructs containing the CDS alone are sufficient for the accumulation of their encoded protein at MT minus ends”, which includes the reference to PMID: 27422905 (ref. n. **74**).

- The sentence on line 118-119 should be edited for precision, since the affirmation is true only for the RNAs tested by RNA FISH.

We have edited the text as follows to highlight that the conclusions refer to RNAs analyzed by smFISH (now lines **112-119**):

*“With this analysis, we confirmed that all basal RNAs analyzed were affected by lack of *Khc* or *aTm1* (Fig. 2D). To check whether the observed changes in RNA localization were specific to basal RNAs, we analyzed the localization pattern of four previously validated apical RNAs (*crb*, *msps*, *qtc*, *CG33129*) in the same mutant backgrounds. In contrast to basal RNAs, none of the apical RNAs analyzed was affected by disruption of kinesin-1-mediated RNA transport (Supplementary Fig. 3, Fig. 2D), indicating that regulators of MT plus end-directed RNA transport specifically control basal RNA localization. These results show that kinesin-1 and *aTm1* are specifically responsible for basal RNA localization in the FE.”*

Reviewer #2 (Remarks to the Author):

In this manuscript, Cassella and Ephrussi uncover mechanisms that underlie RNA localization across the apicobasal axis of follicular epithelia in *Drosophila* embryos. Overall, this manuscript details exciting new findings, and the data presented and conclusions drawn are generally sound. I have only a few comments to make that may improve the manuscript.

We thank the reviewer for appreciating the novelty of our findings and the quality of our data and conclusions.

MAJOR COMMENTS

1. The authors make repeated use of mosaic RNAi lines. These are incredibly powerful and are well-presented. The authors ask how the localization of endogenous and reporter transcripts changes in RNAi cells compared to neighboring unaffected cells. The metric reported for these experiments is always a proportional metric (e.g. DoA) in which signal from apical and basal regions is compared. It could also be useful to know whether the overall number of transcript molecules (or at least amount of smFISH fluorescent signal) is different between the RNAi and unaffected cells. Put another way, is the overall expression/stability of the RNA changing in addition to its localization, or is it the same amount of RNA just ending up at different subcellular locations?

We thank the reviewer for their comment. We now have quantified the total smFISH signal (apical signal + basal signal) for RNAs showing mislocalization phenotypes in all mosaic conditions analyzed (wt vs. KD cells). The data quantification has been added to **Supplementary Figure 2C-D**, the statistical testing is described in Materials and Methods (lines **593-598**), and the results are discussed in the Results section (lines **143-147**, and **180-185**).

Our results show that the only RNA whose amount significantly decreased in RNAi cells compared to wt cells was *CG33129*, upon *Dhc* RNAi. However, the signal intensity of *CG33129* in *Dhc* RNAi cells was considerably higher than its basal intensity in wt cells (highlighted in **Figure for Reviewers 1**), suggesting that degradation of *CG33129* RNA takes place after its change in localization.

It should be also noted that the apical+basal signal detected in the smFISH channel of both *Bsg25D* and *hook* was slightly increased in all CD8-mCherry-marked clones (**Supplementary Figure 2D**) with respect to neighboring unmarked cells. However, this was due to the high laser power applied to image *Bsg25D* and *hook* smFISH probes, which exhibited low labelling efficiency/low transcript abundance; as a result, signal from CD8-mCherry bled through to the smFISH channel, increasing the background mean fluorescence. We have added a note in the legend (lines 52-55) of **Supplementary Figure 2**: “Note that, when imaging the lowly expressed *Bsg25D* and *hook*, the high laser power required to image ATTO-633-labeled smFISH probes resulted in mCherry signal bleed-through that added to the total smFISH signal of mCherry-marked mutant clones, resulting in slightly higher smFISH intensity values in RNAi cells compared to unmarked wild-type cells.”

Figure for Reviewers 1. CG33129 RNA intensity measured in the apical domain (green line) and basal domain (purple line) across *Dhc* RNAi and wt cells. Despite a significant decrease in total smFISH cellular signal, the relative basal intensity of CG33129 RNA is higher in *Dhc* RNAi cells compared to adjacent wt cells, indicating that RNA degradation is subsequent to a change in its RNA localization.

This may be especially relevant to the frameshift constructs. Are these constructs NMD targets?

Thanks to the use of heat shock-induced expression of *BicD-GFP* constructs in clones, we could quantify the smFISH signal of all three *BicD-GFP* RNAs. We did so by 1) measuring the total smFISH intensity in clones expressing each *BicD-GFP* construct (mCherry+) and in cells where the transgene had not been induced (mCherry-) and considered the latter as background signal; 2) subtracting the background signal measured in mCherry- cells from true smFISH signal in mCherry+ cells. In this way, we could compare the fluorescence intensity of all *BicD-GFP* RNAs. Results show that the total smFISH intensity is similar for all *BicD-GFP* constructs (added as a plot in **Figure 5J**), indicating that none of the RNAs is an NMD target.

We have added the following line in the Results section: “Importantly, the change in localization observed for Frameshift *BicD-GFP* RNAs was not due to RNA degradation, as the smFISH signal intensity of all RNA constructs was comparable (**Fig. 5J**).” (lines 237-238 of the revised Main text).

2. The authors found that RNA encoding the dynein adaptor BicD is apically localized in FE cells. They then ask if RNA encoding other dynein adaptors is similarly apically localized, and whether or not this requires active translation (Fig 5E-G). For one of the RNAs investigated, *hook*, there is reasonable evidence that its apical localization requires ribosome association with transcripts as it is sensitive to puromycin but not cycloheximide. For the other transcript interrogated, though, (*Bsg25D*), the data are much, much weaker. There is no significant difference in the localization of *Bsg25D* transcripts in control, cycloheximide, and puromycin treatments. Yet the text still says “the apical localization of both adaptor RNAs showed sensitivity to Puro but not CHX.” I do not agree with this statement as (1) there is no significant difference between the two treatments, and (2) there is no difference at all between the control and puromycin conditions. This statement should be toned down or removed.

We thank the reviewer for their critical comment. As explained in the original text, smFISH probes that anneal to *Bsg25D* RNA show low signal-to-noise ratio due to low expression of *Bsg25D* RNA, and/or low labeling efficiency of smFISH probes. By using the mean fluorescent intensity, capturing small changes in RNA

localization caused by the relatively short treatment with translational inhibitors is challenging. While a prolonged treatment with the translational inhibitor may exacerbate differences in RNA localization, increasing the time of treatment with the drug would severely disrupt tissue morphology.

Following the reviewer's concern which we in fact shared, we re-analyzed changes in the localization of all adapter RNAs treated with translation inhibitors by following a new approach that allowed us to 1) reduce background signal and 2) quantitatively capture small, yet visible, changes in RNA localization along the A-B axis of the cell. To do so, we divided the apical domain into an apical-cortical and a subapical domain. The apical-cortical domain represents the apical-most 1/3 of the apical domain and is the area where most signal is present; the subapical domain is the region between the apical-cortical region and the nuclear periphery (2/3 of the apical domain) (**Supplementary Figure 2B**). For each RNA, in each condition, we first subtracted the background by adjusting the RNA signal threshold; then, we measured the integrated density of particles above the threshold signal for each apical-cortical, subapical, and basal domain. Finally, we calculated the percentage of RNA signal in each of the 3 domains, and plotted their average values ($n=4$ replicates) (**Figure 6E**).

Results replicate our previous findings about *BicD* and *hook* RNAs, where Puro treatment (and not CHX) was shown to significantly reduce their apical localization. Moreover, we could now confidently detect a similar significant change in *Bsg25D* RNA in Puro- (but not CHX-) treated egg chambers. This highlights how thresholding followed by measuring the integrated density allows for a more precise quantification of RNAs with low signal-to-noise ratio along the A-B axis.

Measuring the integrated density following background subtraction allowed us to increase the low signal-to-noise ratio that characterized both *Bsg25D* and *hook* probes. Moreover, the segmentation of FC into 3 subdomains (instead of only apical and basal halves) allowed us to quantify even relatively small changes induced by the short treatment with translation inhibitors that was necessary to preserve tissue morphology. For this reason, we think that the new approach we applied is more suitable to quantitatively capture small changes in RNA localization visualized by smFISH.

The procedure is now summarized in the revised **Supplementary Figure 2B** and described in the Materials and Methods (lines **617-625**), and the results are described in the Results section of the revised Main text (lines **285-292**) and summarized in **Figure 6E**.

3. If the authors wish to argue more forcefully for the translation-dependent apical localization of *Bsg25D* RNA, perhaps they could employ the frameshifted reporter approach they used for *BicD* in Figure 4. This would also remove the technical hurdle of dealing with the low expression level of endogenous *Bsg25D*.

As explained above (point 2), we have now addressed the issue of low expression levels of endogenous *Bsg25D* RNA by replacing the analysis based on line plots with thresholding and measuring the integrated density in the apical-cortical, subapical, and basal domains. Moreover, as Reviewer#1 pointed out, a previous report has shown that the CDS of *Bsg25D* is sufficient to target *Bsg25D* protein to MT minus ends (PMID: 27422905). This finding, together with our improved analysis of translation inhibitor-treated ovaries, strengthens our hypothesis of a translation-dependent mechanism for the localization of *Bsg25D* RNA to MT minus ends.

MINOR COMMENTS

1. For *BicD* RNA, the authors say that "the first peptide emerging from the ribosome is the dynein-binding domain." They use this as possible support for a model in which the RNA's apical localization is cotranslational. Is the dynein-binding domain in the other tested dynein adaptors also N-terminal?

The dynein-binding domain of BICD2 (*BicD*) has been thoroughly investigated and corresponds to the N-terminal coiled-coil domain 1/2 (CC1/2) (PMID: 14609947; PMID: 25814576).

Although the molecular interactions of Hook (HOOK1-3) and *Bsg25D* (NIN/NINL) with dynein have been less well characterized, several reports experimentally linked their N-terminal domain to dynein binding:

- Hook proteins are considered dynein-associated cargo adaptors (PMID: 24637326; PMID: 24637327; PMID: 14697201). The crystal structure of the N-terminal Hook domain has been solved recently (PMID: 27482052) and shows that the N-terminal Hook domain (residues 1-239) binds the C-terminal

region of LIC1 (dynein light intermediate chain 1). This interaction has been characterized in more detail in PMID: 29515126.

- NIN/NINL proteins were recently found to be novel dynein-activating adaptors (PMID: 28718761), with NIN interacting with LIC1 (PMID: 30615611). The interaction with LIC1 was mapped to the first pair of EF domains located in the N-terminal portion of NIN (residues 1-87) (PMID: 33173051)

We discuss the common feature of an N-terminal dynein binding domain in BicD, Hook, and Bsg25D in the Discussion section: “*Puromycin causes the disassembly of the translational machinery and the release of the N-terminal peptides emerging from ribosomes. As the N-terminal portion of these adaptors binds dynein or dynactin subunits^{33,75–78}, we propose that the apical localization of BicD, hook, and Bsg25D depends on the co-translational association between dynein components and nascent adaptors at cortical dynein foci (Fig. 7B).*” (lines 360-364 of the revised Main text) and have updated the references supporting the statement.

2. The schematics alongside some supplementary figures (e.g. Fig S2A, S4A) are very helpful for understanding the experiment. Perhaps it could be considered to include them in their analogous places in the main figures.

In the revised manuscript, we include the schematics in the main figures (**Figure 2A-C** and **Figure 4A,B**).

3. Similarly, the anterior to posterior line across the bottom of the plots in figure S6B is very helpful. Perhaps an analogous addition could be made to the plots in 4E and 5F (in this case apical to basal) to make what is being shown more clear.

In the revised manuscript, we have added an apical-to-basal line to help readers understand the signal distribution in the plots shown in **Figure 5E** and **Figure 5H** (former Figure 4E and Figure 4H, respectively). For greater clarity, we have also replaced the line plot corresponding to the analysis of the localization of *hook* and *Bsg25D* RNAs in translation inhibition experiments (former Figure 5F) with the measurement of the integrated density in the apical-cortical, subapical, and basal domains (see above) that are reported in the x axis label (new **Figure 6E**).

Reviewer #3 (Remarks to the Author):

Cassella and Ephrussi present a considered study that provides a significant advance in our understanding of mechanisms of RNA localization in a rarely-studied system, the follicular epithelium (FE) of *Drosophila* ovaries. The authors first identified a novel pool of apically and basally localized RNAs in the FE using laser-capture microdissection followed by RNAseq analysis of the microdissected tissue. They conducted a number of QC steps such as filtering out likely contaminants and instituting a fairly rigorous threshold of statistical significance, along with validation of some identified RNAs using smFISH. They then tested how knockdown or removal of specific components of transport various machineries affected apical or basal localization of RNAs, and identified three main pathways: 1) an "always-on" dynein- and egalitarian-dependent apical pathway, 2) a kinesin-1-dependent basal pathway that overrides the first pathway for basally localizing RNAs, and 3) a dynein-dependent, translation-dependent, egalitarian-independent pathway that mediates transport of RNAs encoding dynein adaptor proteins such as BicD. Missing from this work, however, is consideration of the functional consequences of these localization pathways. Inclusion of such data should be required for publication in *Nature Communications*.

We thank the reviewer for appreciating our work and considering that it represents a significant advance in our understanding of mechanisms of RNA localization. We anticipate that our work will provide a strong foundation for future in-depth investigation of the functional consequences of the localization pathways.

Other Concerns:

1. The images of the microdissected tissue (lower right in Fig. 1A) are unnecessary. This reviewer recommends moving them to a supplementary figure or deleting them, as they don't illustrate any particular scientific point.

As suggested by the reviewer, we have removed the LCM figures of the original **Figure 1** from the manuscript.

2. A supplementary figure should be included to illustrate how the ROIs are drawn for calculation of Degree of

Apicality (DoA), as it's unclear whether the ROIs are essentially dividing the cell in half (minus the nuclei) or if they are drawn closer to the schematic shown in Fig. S1A, closer to the edges of the cells. Given the patterns of localization, those two different methods would likely lead to different calculations, which would impact reproducibility.

The scheme in **Supplementary Figure 1A** refers to fragment microdissection with LCM, where we purposely aimed at excluding both the nuclei (to avoid contamination from transcriptional foci) and peri-membrane domains (to reduce fragment contamination from neighboring tissues). For all microscopy image analysis, unless specifically noted, we have drawn ROIs dividing the cells in half, including peri-membrane domains and excluding nuclei.

Importantly, the exclusion of peri-membrane domains in LCM fragments may have caused the emergence of false-negative hits in the RNA-seq differential gene expression analysis. Indeed, this is exemplified by *Dhc* RNA, which was characterized by bright peri-membrane smFISH foci, but was not found to be significantly apically enriched in our RNA-seq analysis (**Supplementary Data 1**, see below). Due to the inherent high degree of contamination that characterizes the LCM technique, we strove to reduce the proportion of false positive hits (tissue contaminants) by excluding peri-membrane domains when cutting; on the other hand, this likely resulted in an increase in the proportion of false negative hits (true peri-membrane-localizing RNAs). Nevertheless, the apical vs. basal abundancy (log₂FC) measured with smFISH signal quantification was shown to correlate well with that measured with RNA-seq (**Figure 1D**). This shows that, with the exception of peri-membrane-localizing RNAs such as *Dhc*, the segmentation strategy used to measure the smFISH signal reflects the RNA-seq data.

Following the reviewer's suggestion, we have now included in the revised **Supplementary Figure 2** schematics depicting the two strategies we applied to measure signal intensity. **Supplementary Figure 2A** refers to all experiments where the Degree of Apicality was measured. **Supplementary Figure 2B** refers to the strategy applied to analyze changes in the localization of *BicD*, *hook*, and *Bsg25D* RNAs upon treatment with translation inhibitors.

3. As depicted in Fig. 1C, *Dlic* and *Imp* do not appear to be strongly apical while *AdipoR* and *Rtn1* do not appear to be strongly basal.

To assess more quantitatively the smFISH validation of RNA-seq results we have now quantified the fold change of apical and basal smFISH signal of each RNA in wild-type cells and plotted these values with the respective RNA-seq log₂FC (**Figure 1D**). This shows a good correlation between the two methodologies (adjusted R²: 0.74; p-value: 5.4e-06).

On the basis of our results, we can distinguish two categories of RNA localization. “*Sensu stricto*” localizing RNAs, such as *BicD* (apical) or *CG3308* (basal), are RNAs found almost exclusively in one of the two domains. Enriched RNAs, such as *Dlic* (apical) or *AdipoR* (basal) are those RNAs which are present in both domains but show an enrichment in one domain. There is an inherent variability in the “strength” of RNA localization that might result from RNA-specific mechanistic regulation. In our study we have found that the dynein machinery controls apical RNA localization, while the kinesin-1 machinery controls basal RNA localization. However, additional layers of regulation modulating RNA localization “strength” or association with specific cellular structures could eventually lead to different and RNA-specific localization patterns.

4. It's somewhat concerning given the pattern of localization in Fig. 4J that *Dhc64C* RNA isn't found in the list of apical RNAs. This should be addressed in the text (perhaps the statistics were not sufficiently consistent?).

As mentioned above (point 2), *Dhc64C* was above the significance level in our RNA-seq analysis (FDR>0.1), despite the fact that apical replicates had on average more *Dhc64C* reads than did basal replicates (please see **Figure for Reviewers 2** below). *Dhc64C* RNA as detected by smFISH accumulates strongly in a small number of apical-cortical foci close to the apical membrane. In an attempt to reduce tissue contamination by the neighboring oocyte and muscle tissues, we specifically aimed to exclude peri-membrane domains when performing laser cutting microdissection. Therefore, strong *Dhc64C* RNA foci may well have been excluded from apical fragments or the RNA might have been irreversibly damaged by laser cutting.

We have added this note in the figure caption of **Figure 5** (lines **983-986**): “Importantly, we did not find *Dhc* RNA in our RNA-seq list of apically-enriched RNAs, likely due to the fact that, to minimize contamination from the adjacent tissues, we performed laser cutting excluding cortical regions of the FE where *Dhc* RNA localizes.”

Figure for Reviewers 2. Normalized read counts of *Dhc64C* (*Dhc*) in apical and basal LCM fragments. Points represents read counts in each replicate (n=4).

5. The elucidation of the BicD RNA localization pathway in oocytes (Supplementary Figs. 5 and 6), while interesting, strays from the focus of the manuscript.

Following the reviewer’s comment, we have considerably streamlined this section in the revised Main text (lines **254-268**), and modified **Supplementary Figure 7C** (former Supplementary Figure 6C) (also following the suggestion of Reviewer#1).

Minor Concerns and/or Edits (with line numbers)

Fig. 1A: Scale bars are needed for the images.

As suggested by the Reviewer (point 1), we have removed the LCM images from the figure as we also judged them unnecessary.

Fig. 1A: The use of yellow for the dashed lines around the images of the FE makes it nearly impossible to see. A different color should be used

To facilitate visual interpretation, have replaced the dashed yellow line with a continuous magenta line in all the images highlighting FC clones: **Figure 2A,C; Figure 3A; Figure 4A-C; Figure 6A,B; Supplementary Figure 3A; Supplementary Figure 4A; Supplementary Figure 5A,B.**

Line 22: Change “consists in” to “consists of”.

We have changed “consists in” to “consists of” (now line **22**).

Line 24: Because encoded implies translation, recommend changing “encoded” to “contains”.

We have changed “encoded” to “contain” (now line **24**).

Line 71: Change “consists in” to “consists of”.

We have changed “consisted in” to “consisted of” (now line **71**).

Line 76: Change “aimed at identifying” to “aimed to identify”.

We have changed “aimed at identifying” to “aimed to identify” (now line **76**).

Line 99: “Functionally equivalent” seems overstated. Recommend changing to “similar”.

We have changed “functionally equivalent” to “functionally similar” (now line **99**).

Line 123: Recommend adding “in the absence of kinesin-1”.

We have added “in the absence of kinesin-1”, such that the heading now reads “*Mislocalization of the basal RNA zip in the absence of kinesin-1 depends on Egalitarian*” (now line **149**).

Line 247: The term “adaptor RNAs” makes it sound like the RNA itself is the adaptor. Recommend changing to “adaptor-encoding RNAs”.

In all instances, we have replaced the term “adaptor RNAs” with “adaptor-encoding RNAs” (highlighted in yellow throughout the text).

Line 248: Delete “with the exception of”. Two examples (Nuf and Milton) are not affected and two examples (Bsg25D and hook) are affected. So, Nuf and Milton cannot be described as an exception to a rule.

We have replaced “with the exception of” with “in contrast to” (now line **279**)

Line 765: Change “showed” to “shown”.

We have applied the suggested correction (now line **888**).

We thank the reviewer for all the editorial corrections.

REVIEWERS' COMMENTS

Reviewer #1 (Remarks to the Author):

The authors have done an excellent job in responding to the questions raised in the first review. They have addressed my main concerns.

Reviewer #2 (Remarks to the Author):

The authors have addressed all of my concerns. I congratulate them on an exciting manuscript.

Reviewer #3 (Remarks to the Author):

In this revised manuscript by Cassella and Ephrussi, the authors have responded effectively to most of the concerns detailed in the previous review. Those revisions, as well as those prompted by the other reviewers, have improved the manuscript. The results presented in the revised manuscript are interesting and the data are of high quality. However, the major concern raised in the previous review was the lack of results regarding the functional consequences of RNA localization in the follicular epithelium through the pathways elucidated in this work. Without functional data, this manuscript is not appropriate for publication in Nature Communications.

We sincerely thank all reviewers for their constructive comments and feedback, which have greatly improved the quality of our manuscript. Although we disagree with Reviewer#3 on their final point, we respect their opinion.

REVIEWERS' COMMENTS

Reviewer #1 (Remarks to the Author):

The authors have done an excellent job in responding to the questions raised in the first review. They have addressed my main concerns.

Reviewer #2 (Remarks to the Author):

The authors have addressed all of my concerns. I congratulate them on an exciting manuscript.

Reviewer #3 (Remarks to the Author):

In this revised manuscript by Cassella and Ephrussi, the authors have responded effectively to most of the concerns detailed in the previous review. Those revisions, as well as those prompted by the other reviewers, have improved the manuscript. The results presented in the revised manuscript are interesting and the data are of high quality. However, the major concern raised in the previous review was the lack of results regarding the functional consequences of RNA localization in the follicular epithelium through the pathways elucidated in this work. Without functional data, this manuscript is not appropriate for publication in Nature Communications.